# Pan-Arctic soil element bioavailability estimations

Peter Stimmler[1], Mathias Goeckede[2], Bo Elberling[3], Susan Natali[4], Peter Kuhry[5], Nia Perron[6], Fabrice Lacroix[2], Gustaf Hugelius[5,7], Oliver Sonnentag[6], Jens Strauss[8], Christina Minions[4], Michael Sommer[1], and Jörg Schaller[1*]

[1] Leibniz Centre for Agricultural Landscape Research (ZALF), Müncheberg, Germany
[2] Max Planck Institute for Biogeochemistry, Jena, Germany
[3] Center for Permafrost, Department of Geosciences and Natural Resource Management, University of Copenhagen, Copenhagen, Denmark
[4] Woodwell Climate Research Center, Falmouth, USA
[5] University Stockholm, Stockholm, Sweden
[6] Département de Géographie, Université de Montréal, 1375 Avenue Thérèse-Lavoie-Roux, Montréal, QC H2V 0B3, Canada
[7] Bolin Centre for Climate Research, Stockholm University, Stockholm, Sweden
[8] Alfred Wegener Institute Helmholtz Centre for Polar and Marine Research, Permafrost Research Section, Potsdam, Germany

* Correspondence to: email: Joerg.Schaller@zalf.de

**Abstract.** Arctic soils store large amounts of organic carbon and other elements such as amorphous silicon, silicon, calcium, iron, aluminium, and phosphorous. Global warming is projected to be most pronounced in the Arctic leading to thawing permafrost, which in turn is changing the soil element availability. To project how biogeochemical cycling in Arctic ecosystems will be affected by climate change, there is a need for data on element availability. Here, we analysed the amorphous silicon (ASi) content as solid fraction of the soils, as well as Mehlich III extractions for bioavailability of silicon (Si), calcium (Ca), iron (Fe), phosphorus (P), and aluminium (Al) from 574 soil samples from the circumpolar Arctic region. We show large differences in ASi fraction and Si, Ca, Fe, Al, and P availability among different lithologies and Arctic regions. We summarized these data in pan-Arctic maps of ASi fraction and available Si, Ca, Fe, P, and Al concentrations focussing on the top 100 cm of Arctic soil. Furthermore, we provide values for element availability for the organic and the mineral layer of the seasonally thawing active layer as well as for the uppermost permafrost layer. Our spatially explicit data on differences in the availability of elements between the different lithological classes and regions now and in the future will improve Arctic Earth system models for estimating current and future carbon and nutrient feedbacks under climate change.

# 1 Introduction

Temperatures in northern high latitude region have risen more than twice as fast as the global average within the last decades (IPCC, 2021). This warming leads to thawing of perennially frozen ground known as permafrost (Brown and Romanovsky, 2008; Romanovsky et al., 2010). Frozen conditions prevent organic matter (OM) from microbial degradation and also limits fluvial export of soil-bound nutrients to the sea by runoff (Mann et al., 2022). Thawing of permafrost soils may in turn accelerate global warming by potentially releasing potent greenhouse gases such as carbon dioxide ($CO_2$) and methane ($CH_4$) through soil organic carbon mineralization (Schuur et al., 2015). The frozen ground of the Arctic-boreal regions (hereafter called 'Arctic', but also including subarctic regions) are the largest pool of soil organic carbon worldwide. Approximately, 1014 - 1035 ± max. 194 Pg of organic carbon is stored within the upper 3 m of permafrost region soils (Hugelius et al., 2014; Mishra et al., 2021) (Hugelius et al., 2014; Mishra et al., 2021). To full depth, ca. 1460 - 1600 Pg of organic carbon is stored in the permafrost region (Strauss, 2021), and approximately one third of this is in ice rich Yedoma deposits, exceeding 3 m depth (Fuchs et al., 2018; Strauss et al., 2017b). The Yedoma deposits formed in unglaciated areas of the northern hemisphere during the glacial period, when melt water from glacial areas and eolian processes transported fine sediment to the unglaciated lowlands (Schuur et al., 2013; Strauss, 2021; Strauss et al., 2013). Yedoma deposits are characterized by high carbon and water content. The water is mostly bound in massive ice in ice wedges as well as segregated ice and pore ice in sediment columns (Schirrmeister et al., 2011). Thus, Yedoma soils are highly vulnerable to thawing as the loss of the high ice volume leads to surface subsidence and thermokarst processes, which can accelerate thaw and remobilize deep elemental stocks.

Low temperatures in the Arctic systems slow down biological and chemical processes and preserve OM for millennia (Sher et al., 2005). Due to Arctic warming these processes are accelerated by an increased nutrient and OM mobilization from the permafrost (Salmon et al., 2016). Consequently, OM may become vulnerable to respiration (Hugelius et al., 2020; Strauss et al., 2017b). The temperature and near-surface water content in the Arctic soils have changed rapidly in the last decades and further changes are expected in future (Box et al., 2019). An important effect of Arctic warming is a deepening of the seasonally-thawed active layer and related thermokarst processes, which may lead to a mobilization of nutrients from permafrost soil layers (Abbott et al., 2015). Additionally, increased temperatures in the Arctic regions may accelerate weathering, potentially enhancing nutrient availability in terrestrial Arctic ecosystems and export to freshwater systems, and finally to the nearshore zone and sea (Goldman et al., 2013). As the Arctic features a mineral composition of the soils that is different from many other global soils (Monhonval et al., 2021), the availability (for microbes and plants) of elements in Arctic soils may differ as well. Yedoma deposits, for example, are important pools of OM in the Arctic. Because Yedoma deposits include materials transported from nearby mountains, the mineral compositions of these Yedoma deposits depend on the original geology of the mountains (Schirrmeister et al., 2011). Further, sediments of marine origin are often rich in available calcium (Ca), phosphorus (P) and silicic acid (Si), while magmatic rocks such as granite or basalt contain large amounts of Si, iron (Fe), aluminum (Al), and P. The complex mineral composition of fluvial and marine sediments is reflected in the element availability of the soils formed from these deposits. The availability of these elements in soils is the complex product of soil genesis, nutrient release and plant cycling. The soil properties and the vegetation type interact in terms

of *Sphagno-Eriophorum vaginati* potentially lead to Si accumulation in the uppermost soil layer of the moist acidic
tundra, whereas in the moist non-acidic tundra the *Dryado integrifoliae-Caricetum* may lead to an accumulation of
Ca in the uppermost soil layer (Walker et al., 2001). Further, external inputs, e.g. by fluvial transport in Yedoma
regions, may alter soil genesis and element availability (Monhonval et al., 2021; Strauss et al., 2017a). These processes
depend on physical and chemical conditions including temperature, water content and pH. In this context, Si, Ca, Fe,
Al or P are bound in or on mineral phases and are released via enzymatic activity or weathering. Ongoing Arctic
climate warming now threatens to disturb the pools that have equilibrated to conditions characteristic for the past
millennia.
Nutrient availability (easily available share of elements for potential uptake in plants within short time span e.g. days,
not month) is important to meet the plants requirements in terms of nutrition as only optimal nutrition will result in
high biomass production. The availability of elements such as P, Fe, Ca, and Si in soils are known controls for soil
OM respiration (Weil and Brady, 2017). A release of inorganic nutrients such as P or Si can lead to increased
greenhouse gas production and potentially to further export of these elements to the fresh and seawaters. In marine
systems, P, Fe, Ca, and Si are well-known to control carbon (C) fixation in terms of algae biomass productivity. In
terrestrial systems, P availability is positively related to Si (Schaller et al., 2019) or its polymers, which mobilize from,
e.g., ASi (Stimmler et al., 2021). The mobilization of P by Si was shown to occur due to competition for binding at
Fe-minerals (Schaller et al., 2019), which tend to strongly bind P under condition of low Si availability (Gérard, 2016).
Contrary to Si, Ca binds P by calcium phosphate co-precipitation with calcium carbonate, at least under high soil pH
conditions (Otsuki and Wetzel, 1972). Like P, OM is also binding to Ca, Fe and Al-phases (Kaiser and Zech, 1997;
Wiseman and Püttmann, 2006) but being mobilized from those phases by Si (Hömberg et al., 2020). If the Fe
availability in soils is low, the binding of P may be related to Al-minerals (Eriksson et al., 2015).
Despite the important role of soil elements in driving soil and ecosystem processes and the potential for rapid changes
in the Arctic due to climate change, the spatial distribution of elemental stocks (beyond C, N) is not well understood.
An ecologically based classification of soil Ca concentrations was proposed by (Walker et al., 2001), differentiating
between a Ca rich non-acidic tundra and a Ca poor acidic tundra based on differences in vegetation types for the
Alaskan Arctic region. This classification system was further used to estimate pan-Arctic soil OM stocks (Hugelius et
al., 2014), which proved to be a useful approach as vegetation is tightly connected to OM stocks (Quideau et al., 2001).
Based on the work of (Hugelius et al., 2014), (Alfredsson et al., 2016) related vegetation cover to ASi concentration
to scale up Arctic ASi stocks. However, in contrast to OM stocks being clearly related to vegetation (Hugelius et al.,
2014), the vegetation might have an effect on mineral availability in soils (Villani et al., 2022). It is known that soils
dominated by sedges may become enriched in available Si, whereas soils dominated by shrub vegetation may become
enriched in available Ca (Mauclet et al., 2022). Climate change driven alterations in vegetation communities may lead
to changed element availabilities in Arctic soils. Other elements like P and Ca are cycled by plant intensively and by
this becoming enriched in the uppermost soil layer (Jobbágy and Jackson, 2001). This points to the importance of
biogeochemical interactions between vegetation and soil. However, this approach does not account for the inorganic
element pool, dominated by the bedrock geology initially. Therefore, the extrapolation of circum-polar Arctic maps
of element availability for P, Fe, Ca, Al, and Si based on vegetation distribution alone may be associated with high
uncertainties. A much stronger driver of element availability could be parent material and lithology (Alloway, 2013).
In this study, we aim to map pan-Arctic soil element bio-availability (for microbes and plants) by applying a lithology-
based extrapolation of plot level sampling data on nutrient availability. We provide maps for the solid ASi fraction
and available Si, Ca, Fe, P and Al concentrations as these elements have direct effects on OM binding and greenhouse
gas (GHG) emission from the circumpolar Arctic. In addition, these elements, once transported to the marine systems,
may affect primary production by diatoms and coccolithophores, as some of those nutrients are limiting for those algae
and hence limiting $CO_2$ binding due to algae biomass production. Better understanding of element availability is
crucial to reduce uncertainties for reliable modelling of future scenarios on how Arctic system may respond to global
warming.

## 2 Material and Methods

### 2.1 General approach

Based on the Geological Map of the Arctic (Harrison et al., 2011), we estimate the bio-availability and potential mobility of Si, Ca, Fe, Al and P as well as solid ASi fraction in Arctic soils. We analyzed soil samples from organic, mineral and permafrost layers from pan-Arctic sampling campaigns. We used the biological available element concentrations for mineral Si, Ca, Fe, Al, P and solid ASi fraction of certain lithologies to compile pan-Arctic maps covering 7.6x106 km².

### 2.2 Sampling and storage

In total, we analyzed 574 Arctic soil samples from 25 locations taken over a period of one decade (Fig. 1). To ensure a pan-Arctic coverage we analyzed samples from Siberia (222 samples from six locations), North America (115 samples from six locations), Greenland (111 samples from nine locations), Northern Europe (13 samples from one location) and Svalbard (103 samples from three locations) (Fig. 1 and S1, Table S1). We analyzed samples from the thawed near-surface organic layer (252 soil samples, mainly 0-20 cm in depth), mineral layer (208 soil samples, mainly 20 – 50 cm depth)) and permafrost layer (104 soil samples, mainly 50 – 100 cm depth). The sampling designs accounted for the spatial variation of single locations (several samples were taken on ~1 km transects. Repeated sampling of month to decades was not possible. We split the annually thawed active layer in the upper organic layer and the mineral layer below by C content, except for soils where the organic layer corresponds to the active layer. The organic layer contained mainly organic matter (OM) in different mineralization states. The mineral soil layer has variable OM content depending on which soil processes have affected this layer and reaches to the perennially frozen permafrost layer. We took samples using an auger or a spade and stored them frozen until analysis or as described before (Faucherre et al., 2018; Kuhry et al., 2020). Samples consisted of 5-50 g frozen soil. Before analysis, the samples were freeze dried for 48 hours and ground. The freeze drying inhibits thermal degradation of the soil material and is standard for the Mehlich III extraction and alkaline extraction described below.

### 2.3 Extraction and analysis

*Mehlich III extractions for available element concentrations.*

Available concentrations of Si, Ca, Fe, Al and P were quantified using the Mehlich III method (Sims, 1989). The Mehlich III is extracting the silicic acid which is adsorbed to the soil particle surface and the free silicic acid. For the elements Ca, Fe, Al and P the extract is defined as *biological available share of the analysed elements*, in the script labelled as "*available*". This fraction includes the element concentrations solved in the pore water and the fraction adsorbed to organic and inorganic soil particles. Microbes and plants care able to mobilize this adsorbed share of nutrients. We defined the extraction method Mehlich III to reflect this available element concentrations. Briefly, we extracted 0.5 to 5 g of freeze-dried soil using 10 ml g-1 Mehlich III solution (0.015 M $NH_4F$, 0.001 M EDTA, 0.25 M

NH$_4$NO$_3$, 0.00325 M HNO$_3$, 0.2 M HAc). The samples were shaken for 5 min at 200 min-1 and centrifuged for 5 min
at 10.000 x g. Afterwards the supernatant was filtered using a 0.2 µm cellulose acetate filter. The concentration of Si,
Ca, Fe, Al and P was measured by inductive coupled plasma with optical emission spectroscopy (ICP-OES) (Vista-
PRO radial, Varian Medical Systems, Palo Alto, California).
*Alkaline extraction for solid fraction of amorphous silicon.*
For extraction of solid amorphous silicon (ASi) fraction an alkaline extraction was used (DeMaster, 1981), extracting
ASi from 30 mg of freeze-dried soil using 40 ml 0.1 M Na$_2$CO$_3$ solution at 85°C for 5 h. After 1 h, 3 h and 5 h the
suspension was mixed, and 10 ml of the supernatant was subsampled, filtered by a 0.2 µm cellulose acetate filter and
analyzed by ICP-OES (Vista-PRO radial, Varian Medical Systems, Palo Alto, California). The ASi concentration was
calculated using a linear regression of ASi concentration in solution over time and the intersection with the Y-axis
was used as concentration of available concentration according to (DeMaster, 1981). The Mehlich III extract was used
to determine the available concentrations of the elements and the alkaline extraction was used to determine the pool
of particulate amorphous silicon in the soil. To determine the dry weight (DW) of the samples 0.5-2 g of frozen
material was freeze dried until weight constancy.
**2.4 Statistics**
**2.4.1 Statistics and graphics**
Data were analyzed using the R Studio (R Core Team, 2022). We extracted the original data (lithology, location,
geometry) given for GIS polygons (shape files from the different regions, Greenland, Can_USA, Ice,
N_Europa_Russia) of the Geological Map of the Arctic containing locations. We extracted 14 lithological classes in
total. We matched the soil sampling locations for which we obtained data for element availability by extraction (see
above) with the GIS polygons (geology) by ARCView_GIS_3.2 extensions "Spatial Analyst" command "analysis:
tabulate Areas". The sum of areas with the same map label was extracted by map label "shape area". We considered
only terrestrial areas. For every location, we calculated the mean available element concentrations for ASi, Si, Ca, Fe,
Al and P with bootstrapping (boot=1000) for the organic, mineral and permafrost layer. We calculated quantiles, mean
and standard error using "summarise" from the "dplyr" R package. We clustered available element concentration data
for all locations by lithological class and calculated mean and standard error for organic, mineral and permafrost layer.
The number of samples for each lithological class is given in Fig. 1.
**2.4.2 Element concentration maps**
We used the "Geological Map of the Arctic (1:5 000 000 scale, in the Arctic polar region, north of latitude 60°N")" as
the basis for our maps. We calculated the weighted numeric mean concentration for each element in the first 100 cm
from the soil surface using equation (1). The mean mass fraction (wm) of an element (X) is the sum of the products
of the mass fractions in organic (OL), mineral (ML) and permafrost (PL) layer and the thickness (d) of each layer in
cm divided by 100 cm. We colored the represented area based on the element concentration.
$$w_m\,(X)\,\left[\frac{mg}{g}\right] = \frac{\left(w_{OL}(X)\left[\frac{mg}{g}\right]*d_{OL}\,[m] + w_{ML}(X)\left[\frac{mg}{g}\right]*d_{ML}\,[m] + w_{PL}(X)\left[\frac{mg}{g}\right]*d_{PL}\,[cm]\right)}{100\,[cm]} \qquad \text{(eq. 1)}$$

# 3 Results

## 3.1 Geographical and lithological representation

Our sampling locations represent 13 out of 17 original geographic domains (missing: North Asia and North America, ice, none assigned), defined by the base map (76.5 %) (Table 1, Fig. S1 and Table S2) of the Arctic. The single areas and shares for the maps of Canada/Alaska, Greenland and North Europe/Russia are given in Table S3. Our data represent 17 periods of the Geological Map of the Arctic. The age ranged between 2.6 and 2,500 mya. The number of samples per age code are shown in Fig S2. Our data represent 14 lithological classes of the "Geological Map of the Arctic" (Table S4). These 14 lithological classes represent $7.63\times10^{12}$ m² out of $1.57\times10^{13}$ m² (48.49 % of the area represented by the Geological Map of the Arctic, including ice sheets). Sediments cover $1.03\times10^{13}$ m² of the Arctic. Our data represent sedimentary classes that cover $6.77\times10^{12}$ m² (65.9 % of the Arctic sediment cover) (Fig. S3). In total $3.68\times10^{11}$ m² of $7.37\times10^{11}$ m² (49.9 %) of Yedoma deposits were represented (Fig. S4). The 14 lithological classes can be observed in the igneous type (extrusive: mafic. class 1, n=26), type unclassified (Metamorphic undivided: class 2, n=21) and the sedimentary type (Carbonate: class 3, n=24; class 4, n=58; class 5, n=64; Clastic: shallow marine: class 6, n=13; Clastic: deltaic and nearshore: class 7, n=68; Sedimentary: undivided: class 8, n=38; class 9, n=39; Clastic: shallow marine: class 10, n=91; class 11, n=60; Sedimentary and/or volcanic: undivided: class 12, n=21; and Slope and deep water: class 13, n=43; class 14, n=8).

**Table 1**: Coverage of the areas of geographical domain, epochs, represented area, lithological class, sediments and
Yedoma deposition by our data. First column lists original parameters given by the Geological Map of the Arctic
(Harrison et al., 2011) and Yedoma deposits (Strauss et al., 2021). The column "Represented" gives absolute numbers
for chronological or lithological classes extrapolated by this study. The represented area is the share of the entire area
of the Arctic according to the Geological map used in this study for the listed parameters.

| Parameter | Represented | Explanation | Example |
|---|---|---|---|
| **Geographic domain** | 13 (76.5%) | "Phanerozoic regions are based on major physiographic features of the Arctic" (Harrison et al., 2011) | Interior western Alaska |
| **Epochs** | 17 (2.6 – 2,500 mya) | "Standardization of map-unit attributes has been facilitated by the International Stratigraphic Chart (August 2009 version) published by the International Commission on Stratigraphy (ICS)" (Harrison et al., 2011) | Neogene (23.0 - 2.6 Ma) |
| **Represented area** | $7.63 \times 10^{12}$ m² (43.03%) | Area of the Geological Map of the Arctic (Harrison et al., 2011) containing own data to element concentrations (Fig. 3-8). | |
| **Lithological class** | 14 | Specification and examples of rock type | Lithological class 2: Gneiss, migmatite; reworked amphibolite and granulite facies rocks |
| **Sediments** | $6.77 \times 10^{12}$ m² (65.9%) | Areas with lithological classes of the sedimentary type | Lithological class 7: Sandstone, siltstone, shale, coal; plant fossils; metamorphic grade not identified |
| **Yedoma deposition** | $3.68 \times 10^{11}$ m² (49.9%) | Areas that contain Yedoma deposits defined by Strauss et al. (2021b) | |



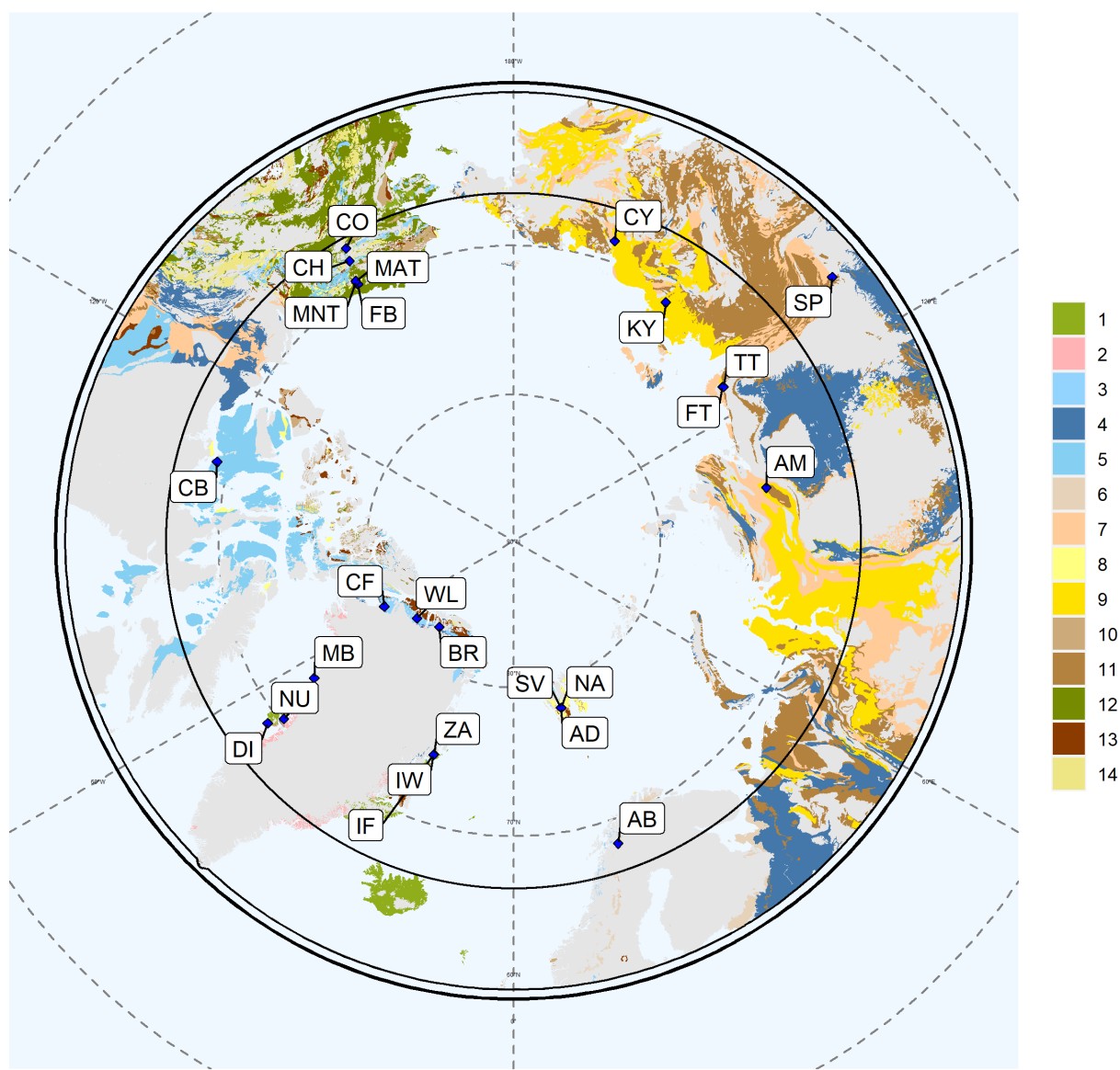



Fig. 1: Map of represented lithologies. The Arctic Circle (66.6°N) is included as a black circle. Each color represents a bedrock lithology: 1: Basalt, olivine basalt, tholeiite, alkali basalt, basanite, pillow basalt, flood basalt (n=26); 2: Gneiss, migmatite; reworked amphibolite and granulite facies rocks (n=11); 3: Limestone, dolostone, shale, evaporites, chalk; carbonate reefs or metamorphosed equivalent (n=24); 4: Limestone, dolostone, shale, evaporites, chalk; carbonate reefs; metamorphic grade not identified (n=58); 5: Limestone, dolostone, shale, evaporites, chalk; carbonate reefs (n=64); 6: Quartz sandstone, siltstone, claystone, limestone, dolostone, conglomerate, tillite (n=13); 7: Sandstone, siltstone, shale, coal; plant fossils; metamorphic grade not identified (n=68); 8: Sandstone, siltstone, shale, limestone (n=38); 9: Sandstone, siltstone, shale, limestone; metamorphic grade not identified (n=39); 10:

Sandstone, siltstone, shale; marine fossils (n=91); 11: Sandstone, siltstone, shale; marine fossils; metamorphic grade
not identified (n=60); 12: Sedimentary and/or volcanic rock: undivided (n=21); 13: Shale, chert, iron-formation,
greywacke, turbidite, argillaceous limestone, matrix-supported conglomerate (n=43); 14: Shale, chert, iron-formation,
greywacke, turbidite, argillaceous limestone, matrix-supported conglomerate or metamorphosed equivalent (n=8).
Grey color means areas of base map that are not represented by our data on element concentrations. Abbreviations for
locations: CH: Alaska, Chandalar; CO: Alaska, Coldfoot; FB: Alaska, Franklin Bluff-Dry; MAT: Alaska, Moist acidic
tundra; MNT: Alaska, Moist non-acidic tundra; CB: Canada, Cambridge Bay; BR: Greenland, Brønlund; CF:
Greenland, Cass Fjord; DI: Greenland, Disko; MB: Greenland, Melville Bay; NU: Greenland, Nussuaq; WL:
Greenland, Warming Land; ZA: Greenland, Zackenberg; IW: Greenland, Zackenberg, Ice Wedge; IF: Greenland,
Zackenberg, Infilling Fan; AM: Russia, Ary-Mas; CY: Russia, Chersky; KY: Russia, Kytalyk; FT: Russia, Lena delta,
first terrace; TT: Russia, Lena Delta, third terrace; SP: Russia, Spasskaya; AB: Sweden, Abisko; AD: Svalbard,
Adventalen; NA: Svalbard, Adventalen; SV: Svalbard. This map is based on the Geological Map of the Arctic
(Harrison et al., 2011).

### 3.2 Element availabilities across lithological classes at 0-1 m depth

The lithological classes differed substantially in their element availabilities (Fig. 2; Fig S5):

- We found a large range in **ASi concentrations** in the Arctic covering values from $0.03\pm0$ mg g$^{-1}$ DW ASi to $6.68\pm1.17$ mg g$^{-1}$ DW ASi. The highest concentrations of ASi were found in basalt and associated rock (class 1: $6.68\pm1.17$ mg g$^{-1}$ DW ASi), Gneiss and associated rock (class 2: $4.11\pm1.24$ mg g$^{-1}$ DW ASi), Sandstone and associated rock (class 9: $2.01\pm0.24$ mg g$^{-1}$ DW ASi. class 10: $2.06\pm0.01$ mg g$^{-1}$ DW ASi). ASi concentrations were lowest in Limestone (class 3: $0.03\pm0$ mg g$^{-1}$ DW ASi) (Fig. 2).

- **Available Si concentrations** were highest in Limestone and associated rock including shale (class 4: $5.65\pm0.78$ mg g$^{-1}$ DW Si), Quartz sandstone (class 6: $6.61\pm1.83$ mg g$^{-1}$ DW Si) and Sandstone (class 7: $5.46\pm0.66$ mg g$^{-1}$ DW Si). Si concentrations were lowest in Limestone and associated rock (class 3: $0.1\pm0.02$ mg g$^{-1}$ DW Si) (Fig. 2). Differences in available Si concentrations in the two classes of Limestone is mainly driven by the presence of shale in one class of the Limestone that acts as source for silicic acid.

- The highest **available Ca concentrations** were observed in Limestone (which consist of $CaCO_3$) and associated rock (class 3: $10.73\pm2.15$ mg g$^{-1}$ DW Ca), Sedimentary and/or volcanic rock (class 12: $8.77\pm0.12$ mg g$^{-1}$ DW Ca) and Sandstone and associated rock (class 8: $8.06\pm0.36$ mg g$^{-1}$ DW Ca). Ca concentrations were lowest in Gneiss (class 2: $0.05\pm0.02$ mg g$^{-1}$ DW Ca) (Fig. 2). The data consists with the expectations of highest Ca availability in Ca containing bedrock.

- **Available Fe concentrations** were highest in shale and associated rock (class 13: $2.93\pm0.45$ mg g$^{-1}$ DW Fe), Limestone (class 4: $2.28\pm0.32$ mg g$^{-1}$ DW Fe) and Quartz sandstone (class 6: $2.49\pm0.69$ mg g$^{-1}$ DW Fe). The lowest Fe concentrations were observed in lithological Limestone and associated rock (class 3: $0.01\pm0.001$ mg g$^{-1}$ DW Fe) (Fig. 2).

- The highest **available concentrations of Al** were observed in to quartz sandstone (class 6: $2.52\pm0.70$ mg g$^{-1}$ DW Al), Sandstone (class 7: $1.63\pm0.20$ mg g$^{-1}$ DW Al) and shale and associated rock (class 13: $1.5\pm0.23$ mg g$^{-1}$ DW Al). The lowest Al concentrations were observed in limestone and associated rock (class 3: $0.02\pm0$ mg g$^{-1}$ DW Al) (Fig. 2).

- High **available P concentrations** were observed in limestone and associated rock (class 4: $0.31\pm0.04$ mg g$^{-1}$ DW P), Sandstone (class 7: $0.19\pm0.02$ mg g$^{-1}$ DW P) and Shale and associated rock (class 14: $0.15\pm0.05$ mg g$^{-1}$ DW P). P concentrations were lowest in Basalt and associated rock (class 1: $0.0116\pm0.002$ mg g-1 DW P) (Fig. 2).

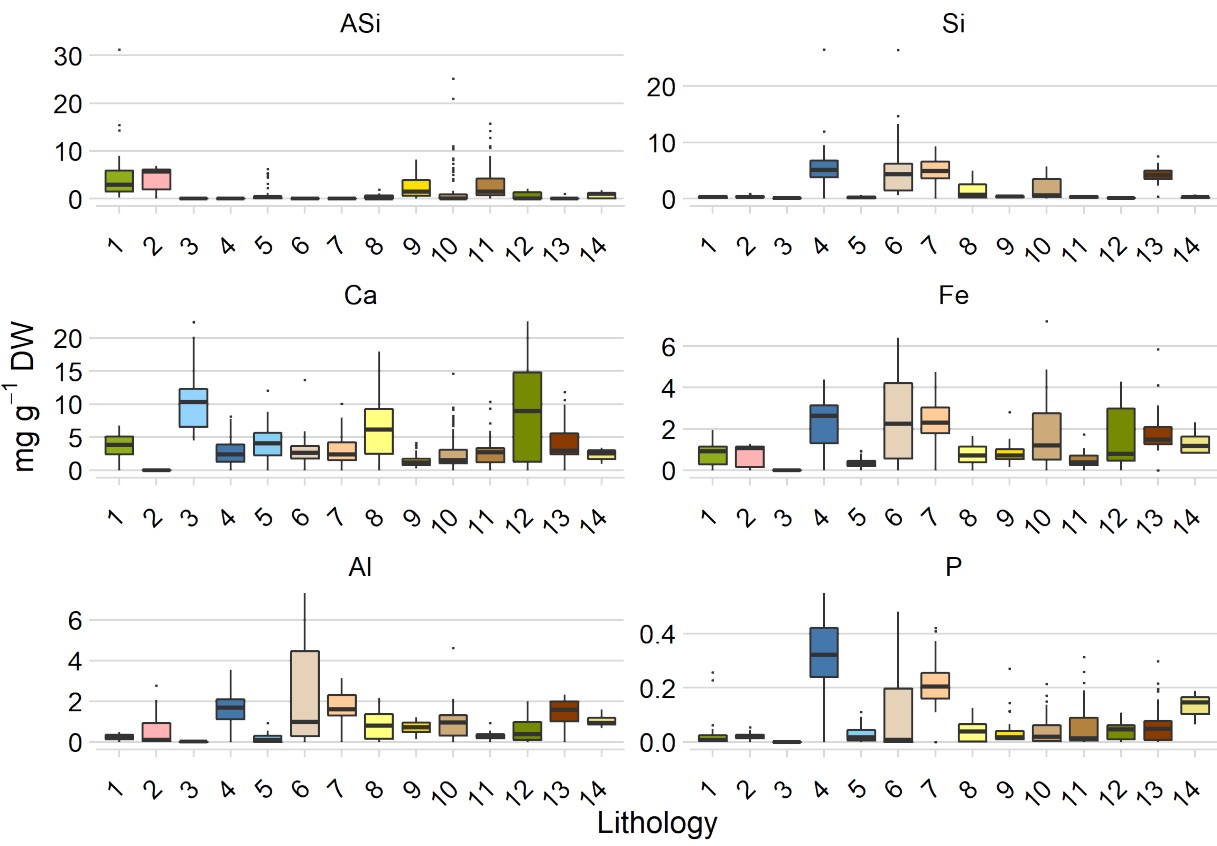

264

Fig. 2: Element concentrations related to lithology. Lithology 1 is igneous, class 2 is metamorphic and the rest are sedimentary, or sedimentary and mixed. Each color represents a bedrock lithology: 1: Basalt, olivine basalt, tholeiite, alkali basalt, basanite, pillow basalt, flood basalt (n=26); 2: Gneiss, migmatite; reworked amphibolite and granulite facies rocks (n=11); 3: Limestone, dolostone, shale, evaporites, chalk; carbonate reefs or metamorphosed equivalent (n=24); 4: Limestone, dolostone, shale, evaporites, chalk; carbonate reefs; metamorphic grade not identified (n=58); 5: Limestone, dolostone, shale, evaporites, chalk; carbonate reefs (n=64); 6: Quartz sandstone, siltstone, claystone, limestone, dolostone, conglomerate, tillite (n=13); 7: Sandstone, siltstone, shale, coal; plant fossils; metamorphic grade not identified (n=68); 8: Sandstone, siltstone, shale, limestone (n=38); 9: Sandstone, siltstone, shale, limestone; metamorphic grade not identified (n=39); 10: Sandstone, siltstone, shale; marine fossils (n=91); 11: Sandstone, siltstone, shale; marine fossils; metamorphic grade not identified (n=60); 12: Sedimentary and/or volcanic rock: undivided (n=21); 13: Shale, chert, iron-formation, greywacke, turbidite, argillaceous limestone, matrix-supported conglomerate (n=43); 14: Shale, chert, iron-formation, greywacke, turbidite, argillaceous limestone, matrix-supported conglomerate or metamorphosed equivalent (n=8). All values are given in mean and standard error. The distribution of the lithological classes is shown in Fig. 1, the assignment to the geographic domain is given in Table S5.

### 3.3 Maps of element concentration in 1 m depth

### 3.3.1 Amorphous silicon in top 1 m

- We found the highest concentrations of ASi located in the Arctic-North Atlantic region (Fig. 3). Here, basalt and Gneiss are dominant (lithological class 1 and 2) and contained concentrations of $4.11 \pm 1.24$ to $6.68 \pm 1.17$ mg ASi g$^{-1}$ DW. Other high concentrations of ASi were found for the Brooks Range (Alaska), Chukotka, Arctic Shelf (eastern Siberia) and the West Siberian Basin. Those soil contained $2.01 \pm 0.24$ mg ASi g$^{-1}$ DW (lithological class 9). The Verkhoyansk-Kolyma region showed a lower concentration of $1.48 \pm 0.16$ mg ASi g$^{-1}$ DW (lithological class 11). We found similar concentrations ($1.24 \pm 0.14$ mg ASi g$^{-1}$ DW; lithological class 5) for the Canadian Shield. We found low concentrations of $0.31 \pm \pm 0.01$ mg ASi g$^{-1}$ DW (lithological class 12) in interior western Alaska and western parts of Brooks Range, Alaska, Chukotka and Arctic Shelf. Increasing active layer depth will potentially release higher ASi concentrations from permafrost soils (Fig. S6) in the Canadian Shield as the concentration in the permafrost layer is $2.80 \pm 2.50$ mg ASi g$^{-1}$ DW (lithological class 5) compared to the $1.24 \pm 0.14$ mg ASi g$^{-1}$ DW in the current active layer (Table S4). A further increase in ASi concentration can be expected for the Arctic, North-Atlantic region by permafrost thaw as the concentration is $8.68 \pm 2.51$ mg ASi g$^{-1}$ DW in the permafrost layer compared to the $4.11 \pm 1.24$ mg ASi g$^{-1}$ DW of the current active layer (lithological class 1) (Fig. S6 Table S4). However, the permafrost layer in Siberia contains lower concentrations of ASi ($0.77 \pm 0.23$ mg ASi g$^{-1}$ DW, lithological class 9 and $1.38 \pm 0.28$ mg ASi g$^{-1}$ DW, lithological class 11) compared to the current active layer with $2.01 \pm 0.24$ mg ASi g$^{-1}$ DW (lithological class 9) and $1.48 \pm 0.16$ mg ASi g$^{-1}$ DW (lithological class 11) probably leading to lower overall ASi concentrations with proceeding thaw. The variability of the ASi data is high for the lithological class 1, 2, 9 and 11.

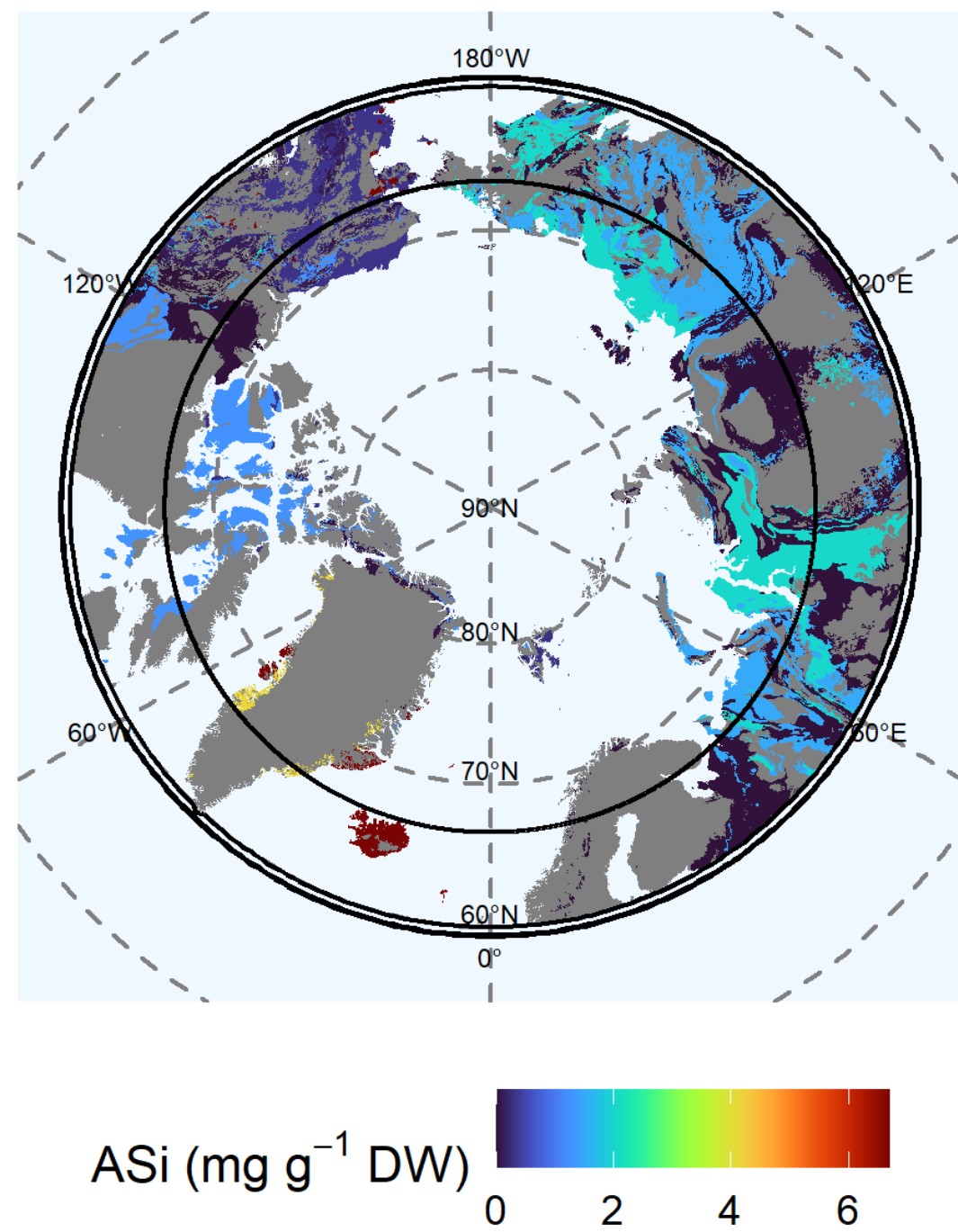

302
Fig. 3: Map of mean concentration of amorphous silicon (ASi) in the top 100 cm of soils. For each lithological class

the mean concentration is shown. Grey shaded areas are not represented by our data.

### 3.3.2 Silicon in 0-1 m depth

Available Si (Mehlich III extractable) (Fig. 4), extracted as silicic acid, showed a different distribution than the solid fraction of ASi (Fig. 3). High available Si concentrations were generally associated with sediments. The available Si extracted by the Mehlich III extraction is water soluble Si plus Si bound to the surface of soil particles (Schaller et al., 2021). We found high concentrations ($5.65 \pm 0.78$ mg Si $g^{-1}$ DW) for lithological class 4, the West Siberian basin and the Siberian plain. Other regions with high available Si concentrations were the East European plain, the Ural Mountains and the Canadian Shield. Another lithological class with high available Si concentrations ($4.51 \pm 0.69$ mg Si $g^{-1}$ DW) is class 13, located in the Innutian Region, North Greenland and in Alaska. In Alaska lithological class 10 with moderate high concentrations of available Si ($2.06 \pm 0.03$ mg Si $g^{-1}$ DW) is also abundant. We found low concentration of available Si ($0.36 \pm 0.05$ mg Si $g^{-1}$ DW, lithological class 9) for Brooks Range, Chukotka, Arctic shelf, the West Siberia Basin and the Siberian Plain. In addition, the Verkhoyansk-Kolyma-Region and the East-European Plain and the Ural Mountains were poor in available Si ($0.39 \pm 0.04$ mg Si $g^{-1}$ DW, lithological class 11). Lowest available concentrations ($0.15 \pm 0.01$ mg Si $g^{-1}$ DW, lithological class 12) were observed in Interior Western Alaska. Increasing thawing depth may potentially increase available Si concentrations in the western Verkhoyansk-Kolyma-Region to the east European Platform as the concentration in the permafrost layer is $6.26 \pm 1.52$ mg Si $g^{-1}$ DW (lithological class 4) compared to the lower available Si concentration of the current active layer with $5.56 \pm 0.78$ mg Si $g^{-1}$DW (lithological class 4) (Figure S6, Table S4). The variability of the data of available Si is high for the lithological class 4, 6 and 7.

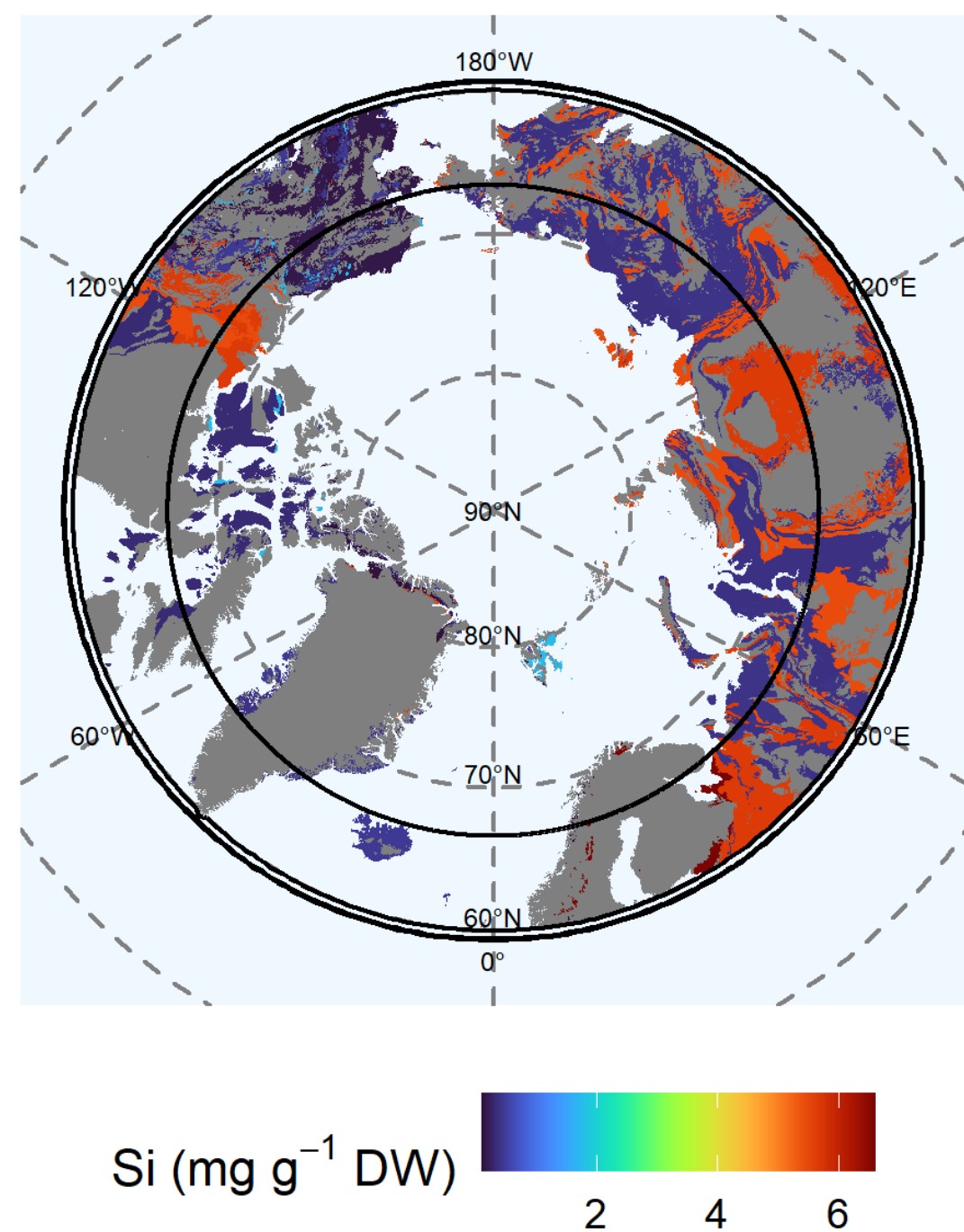

Fig. 4: Map of mean concentration of available silicon (Si) (Mehlich III extractable) for the uppermost 100 cm of

soils. For each lithological class the mean concentration is shown. Blue colors represent low concentrations of

available Si; red colors represent high concentrations. Grey shaded areas are not represented by our data.

### 3.3.3 Calcium in 0-1m depth

The highest available Ca concentrations (Mehlich III extractable) in soils was in limestone and associated rock (class 3: $10.73 \pm 2.15$ mg Ca g$^{-1}$ DW) in North Greenland, Alaska and the Canadian Shield ($3.79 \pm 0.45$ mg Ca g$^{-1}$ DW, lithological class 5) (Fig. 5). Particular limestone shows a high solubility under acidic conditions used in the Mehlich III extraction and by this shows high available Ca concentrations. In addition, supracrustal rocks in Alaska contained very high available concentrations of Ca ($8.77 \pm 0.12$ mg Ca g$^{-1}$ DW, lithological class 12). Mafic rocks in the Arctic North Atlantic region ($3.65 \pm 0.70$ mg Ca g$^{-1}$ DW, lithological class 1) contained moderate available Ca concentrations. We found moderate to low available Ca concentrations ($2.88 \pm 0.32$mg Ca g$^{-1}$ DW, lithological class 11) for the soils of the Verkhoyansk-Kolyma-Region, the East European Plain and the Ural Mountains. Large regions of eastern and western Siberia and the Siberian Plain were poor in available Ca ($1.51 \pm 0.14$ mg Ca g$^{-1}$ DW, lithological class 9; $2.56 \pm 0.34$ mg Ca g$^{-1}$ DW, lithological class 4). The available Ca concentrations of the permafrost layer for Alaska ($10.42 \pm 2.08$ mg Ca g-1 DW, lithological class 12) is higher than in the active layer ($2.93 \pm 0.45$ mg Ca g$^{-1}$ DW) (Fig. S6). In the largest part of Siberia and the Canadian Shield the available Ca concentrations are slightly lower in the permafrost layer with $2.15 \pm 0.96$ mg Ca g$^{-1}$ DW (lithological class 4) and $1.59 \pm 0.32$ mg Ca g$^{-1}$ DW, lithological class 7) than in the active layer with $2.56 \pm 0.34$ mg Ca g$^{-1}$ DW (lithological class 4) and $1.51 \pm 0.14$ mg Ca g$^{-1}$ DW, lithological class 7) (Figure S6, Table S4). The variability of the data of available Ca is high for the lithological class 1, 3, 5 and 6.

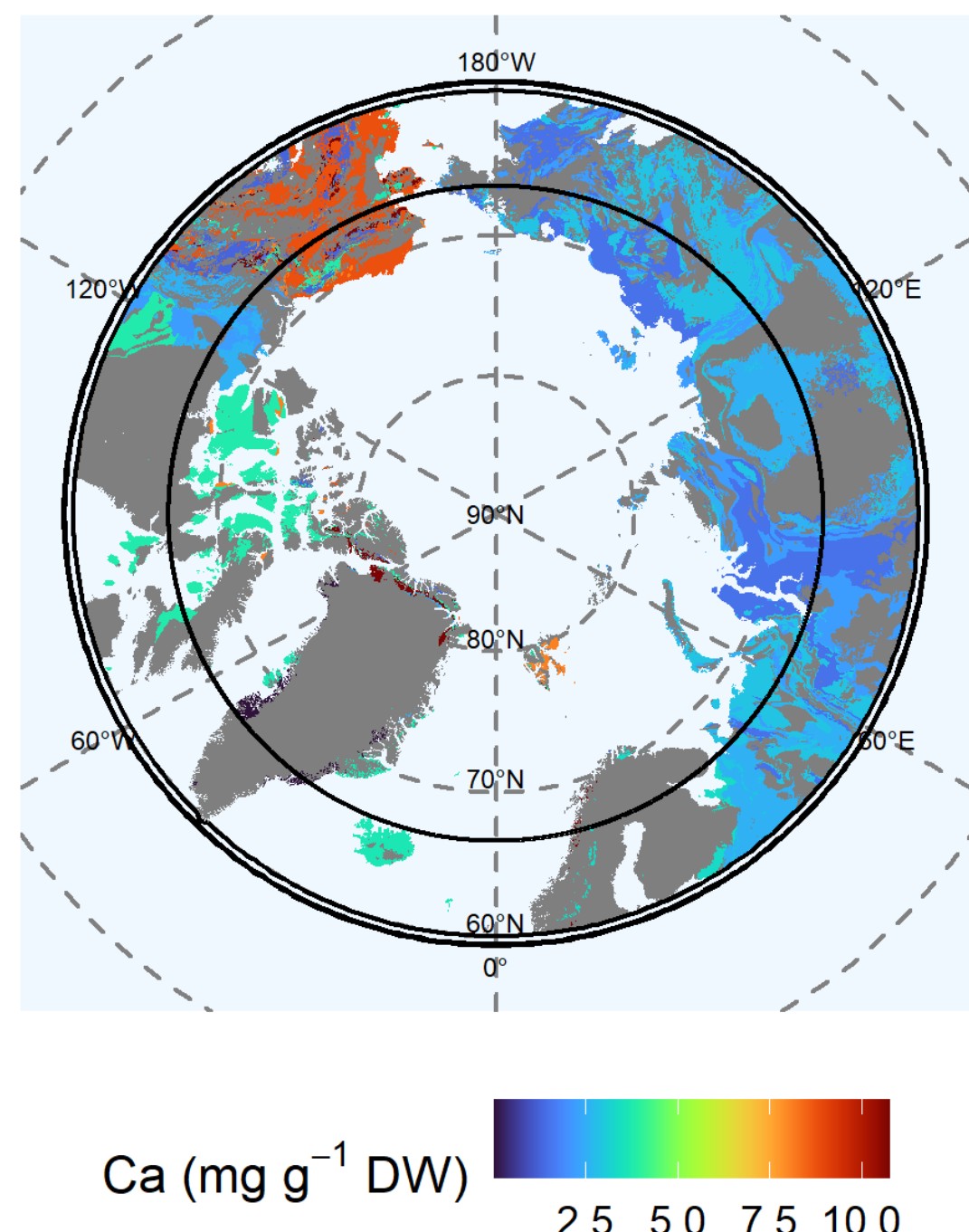

Fig. 5: Map of mean concentration of available calcium (Ca) (Mehlich III extractable) for the uppermost 100 cm of soils. For each lithological class the mean concentration is shown. Blue colors represent low concentrations of Ca, red colors represent high concentrations. Grey shaded areas are not represented by our data.

### 3.3.4 Iron (Fe) in 0-1 m depth

Available Fe concentrations (Mehlich III extractable) were higher in the eastern Arctic, than in the western Arctic (Fig. 6). We found highest concentrations in northern Greenland (lithological class 13 contained $2.93 \pm 0.45$ mg Fe g$^{-1}$ DW). The soils of the lithological class 4 in the western Siberian Basin, Siberian and Canadian plain contained $2.28 \pm 0.32$ mg Fe g$^{-1}$ DW. The Verkhoyansk-Kolyma region showed similar Fe concentrations ($2.21 \pm 0.27$ mg Fe g$^{-1}$ DW, lithological class 7). Moderate to high Fe concentration we found for igneous mafic rocks in Iceland and Greenland ($0.94 \pm 0.18$ mg Fe g$^{-1}$ DW, lithological class 1) and for supracrustal rocks in Alaska ($1.24 \pm 0.14$ mg Fe g$^{-1}$ DW, lithological class 12). The Chukotka region and western Siberia were relatively poor in Fe ($0.83 \pm 0.13$ mg Fe g$^{-1}$ DW, lithological class 9). Eastern Siberia and North Europe contained even lower Fe concentrations ($0.49 \pm 0.04$ mg Fe g$^{-1}$ DW, lithological class 11). Available Fe concentrations in the Canadian Shield were similarly low ($0.41 \pm 0.06$ mg Fe g$^{-1}$ DW, lithological class 12). We expect increasing Fe concentrations at the Canada and Greenland shield due to predicted future thaw of the permafrost layer as the concentration in the permafrost layer ($0.61 \pm 0.15$ mg Fe g$^{-1}$ DW, lithological class 5) and in parts of Alaska ($1.97 \pm 0.3$ mg Fe g$^{-1}$ DW, lithological class 14) is higher compared to the current active layer with ($0.41 \pm 0.05$ mg Fe g$^{-1}$ DW, lithological class 5) and in parts of Alaska ($1.08 \pm 0.38$ mg Fe g$^{-1}$ DW, lithological class 14) (Fig. S7, Table S4). The variability of the data of available Fe is high for the lithological class 2 and 6.

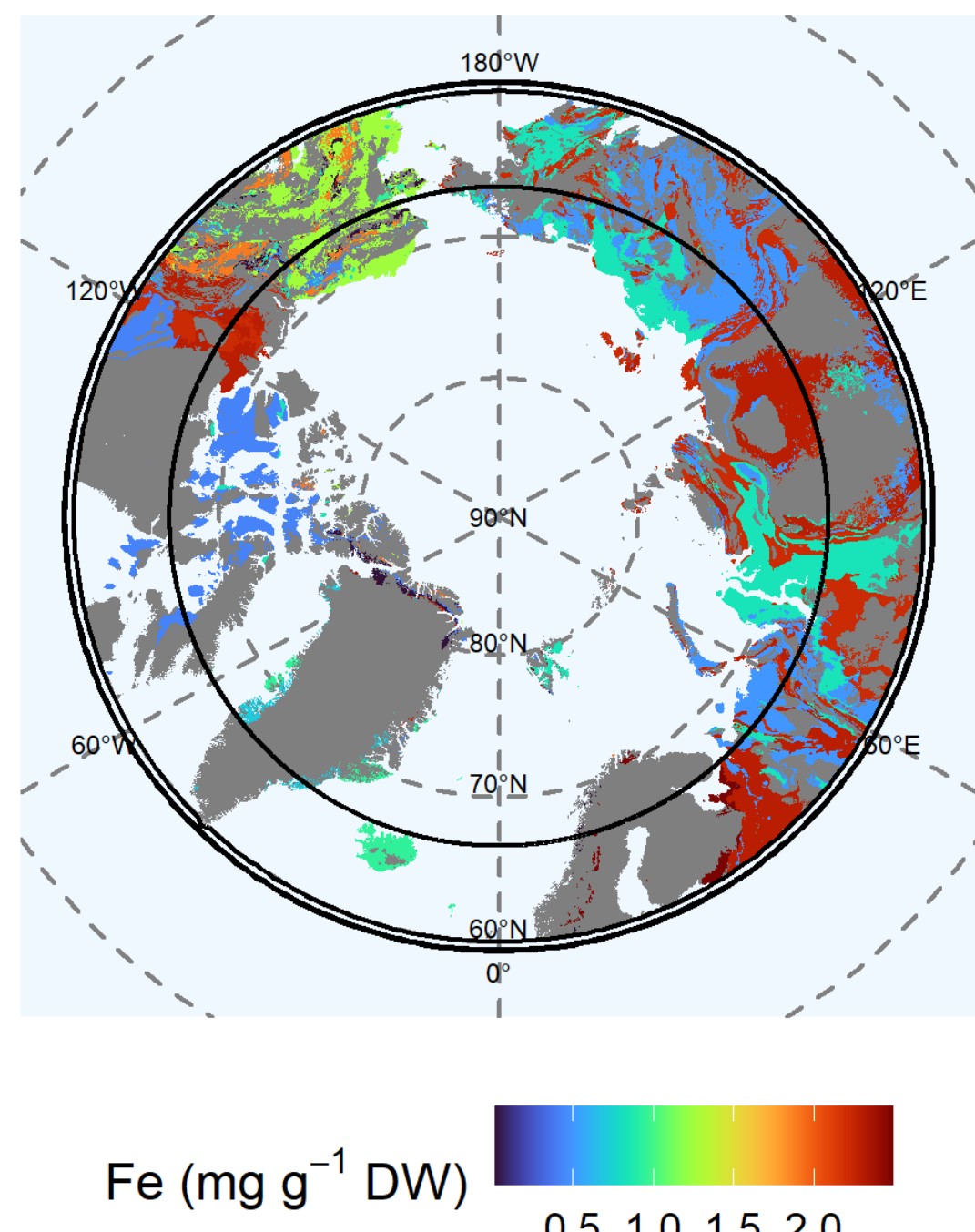

Fig. 6: Map of mean concentration of available iron (Fe) (Mehlich III extractable) for the uppermost 100 cm of soils.

For each lithological class the mean concentration is shown. Blue colors represent low concentrations of Fe, red colors

represent high concentrations Grey shaded areas are not represented by our data.

### 3.3.5 Aluminum in 0-1 m depth

Northern Europe contained highest concentrations of available Al (Mehlich III extractable) ($2.52 \pm 0.7$ mg Al g$^{-1}$ DW, lithological class 6) (Fig. 7). Relative high concentrations of available Al were distributed over Siberia and the Canadian Shield ($1.63 \pm 0.02$ mg Al g$^{-1}$ DW, lithological class 7; $1.57 \pm 0.22$ mg Al g$^{-1}$ DW lithological class 4). Parts of Alaska contained moderate available Al concentrations ($0.94 \pm 0.06$ mg Al g$^{-1}$ DW, lithological class 10; $1.5 \pm 0.23$ mg Al g$^{-1}$ DW, lithological class 13), while areas represented by supracrustal rocks were poor in available Al ($0.47 \pm 0.06$ mg Al g$^{-1}$ DW, lithological class 12). We found relative low concentrations ($0.73 \pm 0.01$ mg g-1 DW) of available Al for Chukotka, eastern and western Siberia observed in lithological class 9. The Verkhoyansk-Kolyma Region and the East European plain showed the lowest available Al concentrations ($0.26 \pm 0.02$ mg Al g$^{-1}$ DW, lithological class 11), together with the Canada plain ($0.21 \pm 0.03$ mg g$^{-1}$ DW Al, lithological class 5). Increasing thawing depth may increase the available Al concentration by the predicted thaw of the permafrost layer in North Europe as the concentration in the permafrost layer is $4.88 \pm 1.02$ mg Al g$^{-1}$ DW (lithological class 6) and across the Greenland and Canadian shield it is $0.3 \pm 0.07$ mg Al g$^{-1}$ DW (lithological class 5) compared to the current active layer with $2.52 \pm 0.7$ mg Al g$^{-1}$ DW (lithological class 6) and across the Greenland and Canadian shield it is $0.21 \pm 0.03$ mg Al g$^{-1}$ DW (lithological class 5) (Fig. S7, Table S4). The variability of the data of available Al is high for the lithological class 2 and 6.

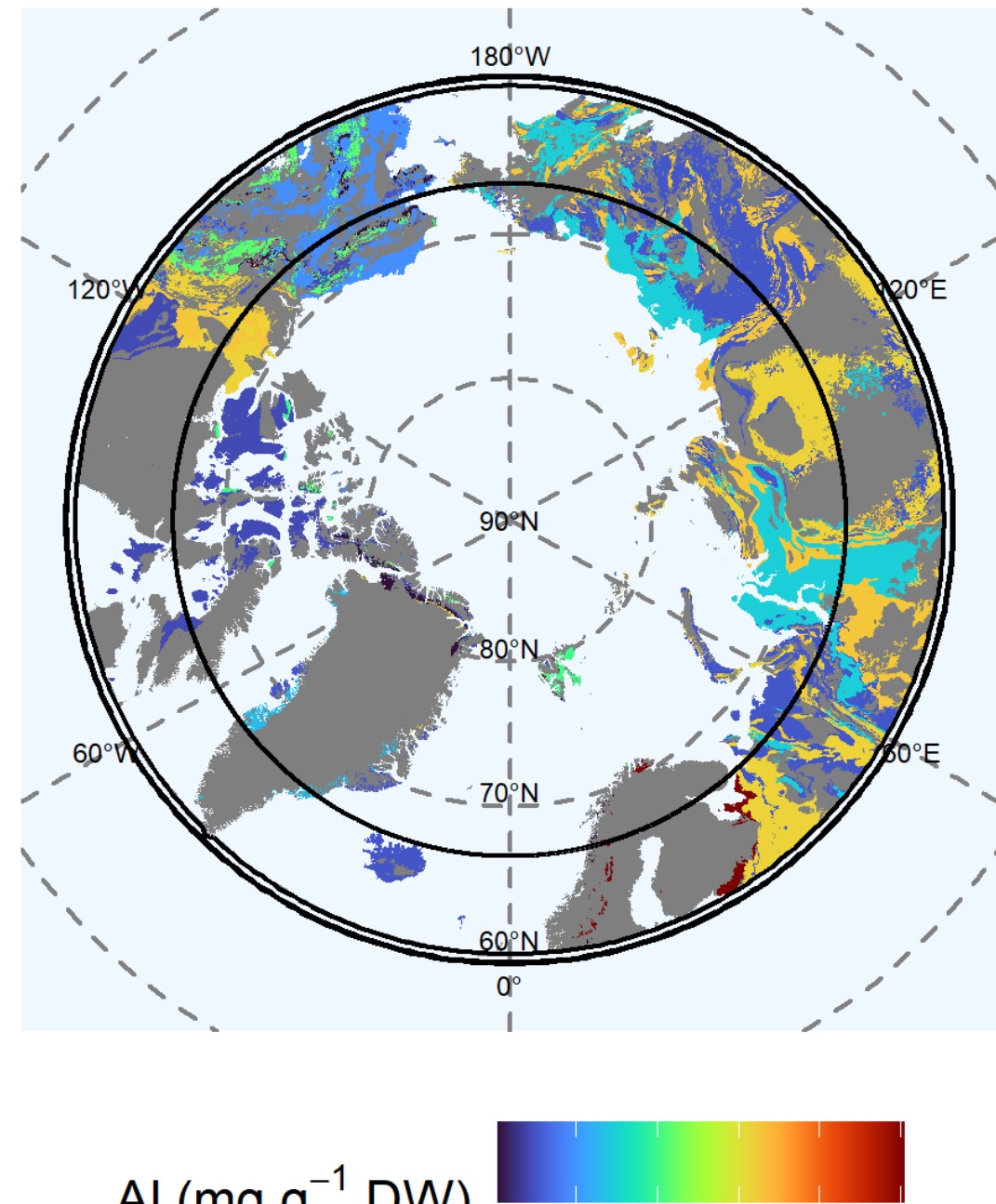

Fig. 7: Map of mean concentration of available aluminium (Al) (Mehlich III extractable) for the uppermost 100 cm of soils. For each lithological class the mean concentration is shown. Blue colors represent low concentrations of Al, red colors represent high concentrations. Grey shaded areas are not represented by our data .

### 3.3.6 Phosphorous in 0-1 m depth

We found the highest available P concentrations (Mehlich III extractable) we found for the West Siberian Basin, the Canadian Shield and the Siberian and East European Plain ($0.306 \pm 0.042$ mg P g$^{-1}$ DW, lithological class 4) (Fig. 8). In the Chukotka region the available P concentrations were $0.189 \pm 0.023$ mg P g$^{-1}$ DW (lithological class 7. We found moderate available P concentration for Northern Europe ($0.123 \pm 0.034$ mg P g$^{-1}$ DW, lithological class 6) and in Alaska ($0.153 \pm 0.054$ mg P g$^{-1}$ DW, lithological class 14). Wide areas of supracrustal rocks in Alaska were poor in available P ($0.024 \pm 0.004$ mg P g$^{-1}$ DW, lithological class 12). The Canadian shield ($0.037 \pm 0.005$ mg P g$^{-1}$ DW, lithological class 5), the Verkhoyansk-Kolyma region, the east European Plain ($0.017 \pm 0.002$ mg P g$^{-1}$ DW, lithological class 11) and the Chukotka region ($0.030 \pm 0.003$ mg P g$^{-1}$ DW, lithological class 9) were poor in P. Due to permafrost thaw we expect increasing available P concentrations the Canadian shield as the available P concentrations in the permafrost layer is $0.06 \pm 0.01$ mg P g$^{-1}$ DW (lithological class 5) compared to the current active layer with $0.04 \pm 0.005$ mg P g$^{-1}$ DW (lithological class 5) (Fig. 7, Table S4). The variability of the data of available P is high for the lithological class 4, 6 and 7.

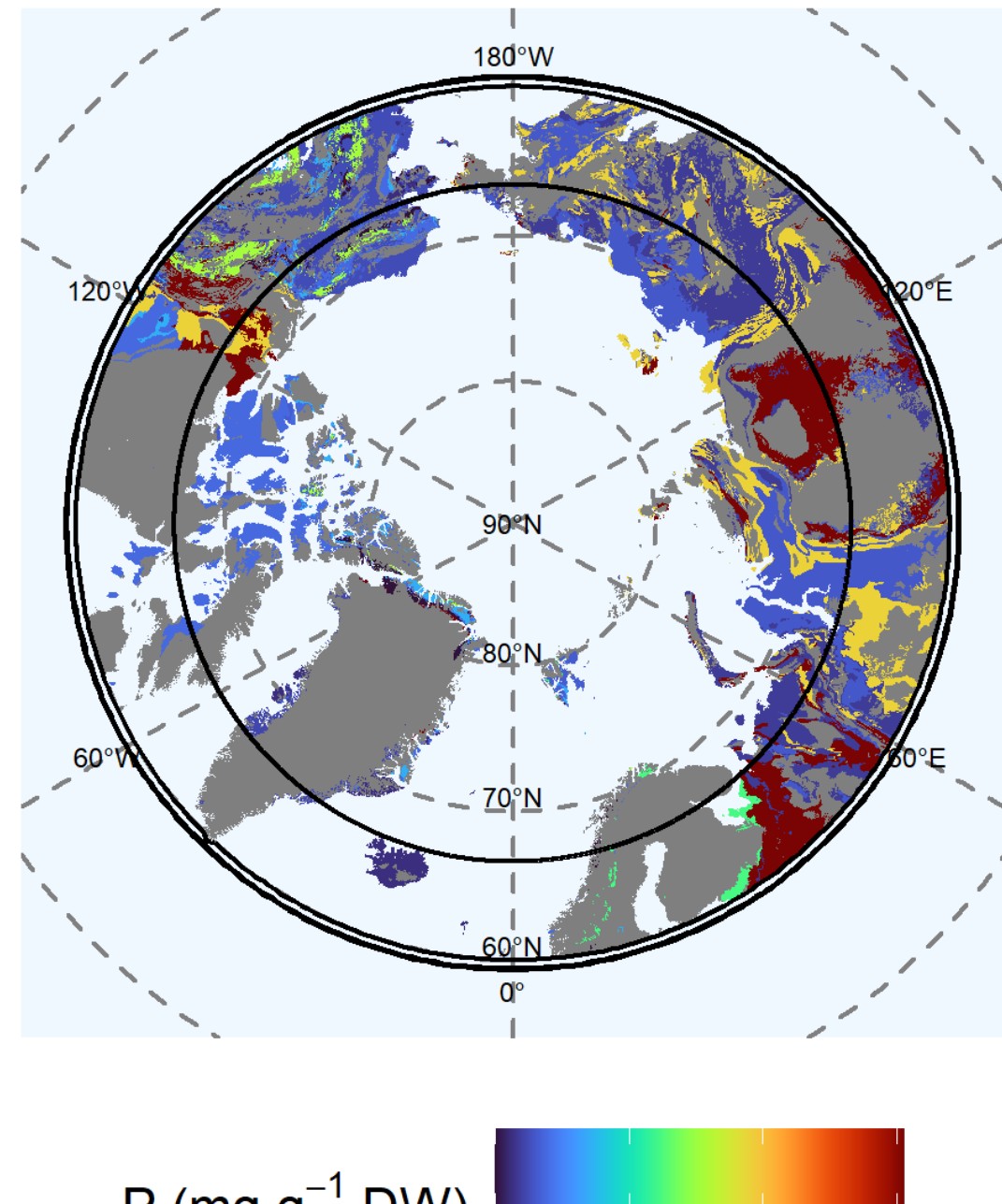

Fig. 8: Map of mean concentration of available phosphorous (P) (Mehlich III extractable) for the uppermost 100 cm

of soils. For each lithological class the mean concentration is shown. Blue colors represent low concentrations of P,

red colors represent high concentrations. Grey shaded areas are not represented by our data .

## 4 Discussion

### 4.1 Element availability in relation to lithology and geography

We found large differences in the availability of all analysed elements between the different lithology classes of the Arctic. The igneous lithological type for example is dominated by alkaline and Ca-rich basaltic rocks from Alaska. Sedimentary rocks covers a wide range of pH, as sediments of diverse origin can contribute to form a sedimentary rock (Schirrmeister et al., 2011). In regions with fluvial deposition, e.g. in Yedoma affected areas, the soil forming material may differ from the underlying bedrock (Kokelj et al., 2017). Limestone sediments for example differ in their content of available Fe and P, depending if their origin is biological (lithological class 4), physical or chemical. Sandstone can contain high available Fe concentrations, too, but it contains available Si as the main element (lithological class 7-8) (Yurchenko et al., 2019). Previously, there was no map existing for availability of Si, Ca, Fe, P, and Al in Arctic soils, and only a map on ASi stocks but not concentrations. Our maps show element concentration available for plants and microbes together. We further show potential changes in element availability by deepening of the active layer, element export by run-off from thawed soil and thermokarst processes.

### 4.2 Relevance of element availability in a dynamic Arctic

Nutrient cycles and limitations were identified as important for improving of high latitude ecosystems estimations vegetation functional parameter like gross primary production (GPP) (Chadburn et al., 2017). The dataset presented in our study could therefore serve as a basis for providing soil nutrient concentrations for biogeochemical models that are capable of considering nutrient limitations in permafrost ecosystems. Our maps cover nearly the half of the Arctic area. The distribution of ASi in Arctic regions was first estimated by (Alfredsson et al., 2016). (Alfredsson et al., 2016) showed maps of ASi stocks (not concentrations) related to vegetation cover, covering 30 soil profiles. Elements like Si and Ca are accumulated by plants, depending on the vegetation type (Jobbágy and Jackson, 2001; Mauclet et al., 2022). By this the effect of current vegetation on element availability in soils is associated with high uncertainties as the vegetation involved in forming the soil ASi pool may be different from the current vegetation. A more appropriate measure of element availability may be parent material and lithology (Alloway, 2013). It was shown, that geochemical element concentration in Arctic permafrost soil allow to distinguish geologies (Reimann and Melezhik, 2001). Consequently, our lithology-based extrapolation of nutrient availability will help to reduce the so far uncertainties in pan-Arctic soil element availability.

Due to deepening of the active layer, as for example at the Canadian Shield and in the North-Atlantic region, our data suggest higher ASi concentrations in the active layer in future, as the concentration in the permafrost layer is higher compared to current active layer (Fig. S6). This higher ASi concentrations may increase P and OM availability (Reithmaier et al., 2017; Schaller et al., 2019) by polysilicic acid competing for binding sites on the surface of minerals subsequently mobilizing both P and OM as ASi is a main source for polysilicic acid, potentially increasing the leaching of both elements to the sea. It was also shown that available Si leads to a release of P from minerals of Arctic soils and increases OM decomposition, increasing soil greenhouse gas release (Schaller et al., 2019; Schaller et al., 2021b).

Available Ca can immobilize OM by cation bridging and by this preserve OM from microbial decomposition (Sowers
et al., 2020). Available Ca is relevant for the mineral formation because it can bind $CO_2$ as $Ca(HCO_3)_2$ in soil with pH
higher than 7 (Dessert et al., 2003; Köhler et al., 2010). The concentration of soluble Ca in Yedoma soils is mainly
driven by thermokarst processes (Monhonval et al., 2022). In permafrost layers, the data presented in Fig. S6 shows
Ca concentrations are in most locations lower than in the current active layers, which is in accordance to other studies
(Kokelj and Burn, 2005). Consequently, a future increase in temperatures may lead to a widespread decrease in
available Ca concentrations in average at 0-1 m depth, especially in the Yedoma regions. Iron minerals are important
electron acceptors under anaerobic conditions and available Fe is essential forko microbial methane production
(Colombo et al., 2018). After being released from rocks by weathering, Al forms amorphous aluminosilicates that
crystalize slowly (Schaller et al., 2021a). The Mehlich III extract contains all soluble forms of $Al(OH)(H_2O)$ that are
bioavailable for organisms, with Al being toxic (Rengel, 2004). Thawing permafrost may be a source for available Al,
especially across Canada, the Greenland shield and Northern Europe. Increasing P availability, as predicted for
Greenland and the Canadian shield (Fig. 7) may for example increase $CO_2$ release to the atmosphere by increasing the
mineralization rates of OM (Street et al., 2018; Yang and Kane, 2021).

**4.3 Importance of element interactions for nutrient availability**

In permafrost layers, the mineralization of OM by microbial activity is negligible due to frozen conditions. Like in
temperate soils, binding of OM on mineral phases can prevent OM from mineralization (Dutta et al., 2006; Mueller et
al., 2015). Mineral phases may bind parts of soil OM reducing the amount of OM for microbial respiration. A large
share of OM may be associated with iron and aluminium oxides/hydroxides. In particular iron minerals may strongly
bind OM, whereby a high stability of stored carbon is likely (Herndon et al., 2017). Thereby the binding between OM
and the minerals is determined by the quantity of minerals that can bind OM (Wiseman and Püttmann, 2006). This
would imply that a higher concentration of available Fe, Al and Ca in Arctic soils due to permafrost thaw may lead to
a lower GHG emission from Arctic soils due to complexation of OM with those elements. Such increase in element
availability binding OM and with this resulting in potentially lower GHG emissions may happen for Alaska (higher
available Ca and available Fe concentration in permafrost layer compared to current active layer), Canadian Shield
and Greenland (higher available Fe and available Al concentration in permafrost layer compared to current active
layer), and North Europe (higher available Al concentration in permafrost layer compared to current active layer) (see
results part). However, lower available Ca concentrations can be expected in large parts of Siberia and the Canadian
Shield, as the concentrations in the permafrost layer are lower compared to the current active layer. Available silicon
however, can potentially mobilize OM from those phases under slight acidic to alkaline and also from oxic to anoxic
conditions, by binding competition of silicic acid with some functional groups of organic material (Hömberg et al.,
2020), potentially increasing GHG emissions. An increase in Si availability upon permafrost thaw can be expected in
the western Verkhoyansk-Kolyma-Region to the east European Platform as the concentration in the permafrost layer
is higher compared to the current active layer (see results part). Available P competes with OM for binding on soil
minerals (Schneider et al., 2010). Such increasing P concentration due to permafrost thaw can be expected for the
Canadian Shield (results part). Based on the differences in element (Si, Fe, Al, Ca and P) available concentration, the
stability of OM differs in Arctic regions, depending on the dominating mineral composition, lithology and element
availability. Also the availability of nutrients (P in this case) is modified by mineral composition. For example, P is
often strongly bound to Fe mineral phases, reducing P availability (Gérard, 2016). Silicic acid however, is able to
mobilize P from strong binding to Fe mineral by competing for binding sites in a wide range of conditions (Schaller
et al., 2019). Unlike Si, Ca binds P by calcium phosphate precipitation at alkaline conditions (Cao and Harris, 2008)
or as calcium carbonate/phosphate co-precipitation (Otsuki and Wetzel, 1972). Under conditions of low Fe availability
in soils, the binding of P may be related to Al-minerals (Eriksson et al., 2015). A lack of available P leads also to a
reduction of the physiological activity of microbes (Walker et al., 2001), thus potentially reducing microbial
respiration of OM. Mobile Si in forms of silicic acid and its polymers may potentially limit the availability of ions
like Fe, Al, or Ca by precipitating those elements in amorphous or crystalline phases (Schaller et al., 2021a). Hence,
the mobilization of elements like Si, Ca, Fe and Al strongly interfere with both P and OM availability and thus
potentially with GHG emissions. To unravel the dominant processes upon permafrost thaw, or which element
mobilization is dominant in terms of OM binding or mobilization and with this affecting GHG emission, future studies
are urgently needed.

**4.4 Transport of elements to the Arctic Ocean**
With the ongoing deepening of the active layer in Arctic soils, an increased leaching of elements and nutrients may
occur (Mann et al., 2022; Sanders et al., 2022), which may substantially impact marine biodiversity and ecosystem
function. We have shown for several regions of the Arctic that there will be regional differences in element
mobilization upon permafrost thaw. For example, increased export of Fe and P, which are the main limiting nutrients
for marine net primary production (NPP); (Zabel and Schulz, 2006), has already contributed to a 30% increase in NPP
in the Arctic Ocean between 1998 and 2009 (Arrigo and van Dijken, 2011). Increased available Fe concentrations at
the depth of 0-1 m in the soils upon permafrost thaw can be expected for soils developed on the Canadian Shield,
Greenland and Alaska, whereas increased P mobilization may occur only in the Canadian Shield, according to the
sites studied within these lithological classes. Silicon and Ca also have a crucial role in marine primary production.
Both elements are components of the inorganic spheres of diatoms (Si) and coccolithophores (Ca), which fix $CO_2$ in
the Arctic Ocean, an important global carbon sink (Krause et al., 2018). At the Arctic Canadian coast, Si inputs led to
an increase of diatoms from 2% to 37% (Terhaar et al., 2021). Diatoms and coccolithophores are the basis of the
marine food chain, and therefore, shifts in their populations may have widespread implications for the marine
ecosystem (Daniels et al., 2018). Permafrost thaw is likely to accelerate inputs of available Si and Ca to Arctic waters.
Increased Si availability in soils can be expected in the western Verkhoyansk-Kolyma-Region to the east European
Platform as the concentration in the permafrost layer is higher compared to the current active layer, (see Results
section). Calcium mobilization may increase or decrease depending on the Arctic region. Increase Ca mobilization
can be expected for Alaska, whereas a slight decrease in Ca mobilization may occur in large parts of Siberia and the
Canadian Shield (see Results section). Yedoma deposits readily leach soluble ions, including Si and Ca, as a result of
thaw degradation (Strauss et al., 2017b). Alaskan soils store huge amounts of available Ca in the mineral layer (see
above) that could be transported to the Beringia Sea with increasing soil degradation, promoting the growth of
coccolithophores. In Siberia, the Lena River could transport large amounts of available Si to the Laptev Sea increasing
the growth of diatoms. The same could happen at the East European plain. In the same way P concentrations in these
regions of the Arctic Sea could rise, too, as P concentrations in the permafrost layer of the Canadian Shield are higher
compared with the current active layer (see results part). During the transport to the ocean the elements may be bound
to soil particles potentially decreasing the further transport or may interact with other elements (see paragraph before)
also potentially affecting the further transport. In summary, in many areas of the Arctic with high available Si, Ca and
P storage, there could be increased inputs to Arctic waters with permafrost thaw potentially increasing $CO_2$ fixation
by marine primary production.

**4.5. Limitations of data and statistics**
Despite sample number of our study being quite high and reflecting a broad range of the pan-Arctic regions, this study
has still some limitations. The density of data points is not homogeneous over the whole area and in some remote
areas the sample number is low. To reduce potential uncertainties and variance in the presented data on available
element concentrations we did bootstrapping for the single layers within the lithological classes. Our data does not
give total element pools, but biological available concentrations.

**5 Data availability**
The data for element availability from all single locations, soil profiles, transects, lithology's, as well as bootstrap data
for location and lithology can be can be downloaded via the open-access MPG repository EDMONT under
https://doi.org/10.17617/3.8KGQUN (Schaller and Goeckede, 2022). During review process, the data is available
under:  https://edmond.mpdl.mpg.de/privateurl.xhtml?token=8cbb0bd8-790f-4719-8cd1-a3df4ff99477  to  allow
corrections based on reviewer comments (Schaller and Goeckede, 2022). The repository contains a readme file ("Read
me.docx"). In this file, all necessary information can be found, including all columns descriptions need to use the data.
The element availability from all single locations, soil profiles, transects, lithology's labelled (loction_samples.txt)
with following parameters: geological map of the Arctic, individual ID of the polygon, official name of the sampling
site, study internal name of the soil sample, soil horizon, coordinates of sampling sites, concentration of alkaline
extractable amorphous silicon (ASi), Mehlich-3 extractable Si, Ca, Fe Al and P, thickness of the layer, original depth
where soil was taken, size of the polygon that contains the sampling site, age code, scientific name of the age, where
bedrock was formed, scientific name of the eon, where bedrock was formed, scientific name of the era, where bedrock
was formed, scientific name of the period, where bedrock was formed, maximal and minimal age of bedrock,
information if lithogenesis was of the supracrustal, sedimentary or igneous type, most common rock types in the
cluster group of the setting, code of metamorphic type, code of domain region, name of tectonic and geographic
domain, as well as name of region within geographic domain.

In the location_bootstrap.txt file the bootstrapped means of concentration of alkaline extractable silicon, Mehlich-3 extractable Si, Ca, Fe Al and P for the organic, mineral active and permafrost layer of the single locations are given in mg/g DW.

In the file "lithology_bootstrap.txt" element concentration for the first 1 m, including organic, mineral and permafrost layer are given as bootstrapped mean with standard deviation for alkaline extractable silicon as well as Mehlich-3 extractable Si, Ca, Fe Al and P.

## 6 Conclusion

Here, we identified large differences in concentrations in available Si, Ca, Fe, P, Al and the solid ASi fraction between different Arctic regions. With the future projected warming of the Arctic and the associated thaw of permafrost soils by deepening the active layer, the available concentration of the elements will change, . Depending on dominance or limitation of certain elements, biogeochemical processes such as OM mineralization may increase or decrease. Moreover, not only microbial processes like OM respiration may be affected by changes in Si, Ca, Fe, P, and Al availability, but also processes such as primary production ($CO_2$ fixation by plants) in terrestrial systems. This could be stabilizing soil OM, but may also trigger elevated biomass production of plants due to increased nutrient supply. In addition, marine systems will receive higher loads of leached elements, which could increase algae biomass production due to larger nutrient transport to the sea. Our spatially data product including the differences in elements availability between the different lithological classes and regions will help improving models of Arctic biogeochemical cycles for estimating future carbon feedback under the predicted climate change.

**Competing interests.** The contact author has declared that neither they nor their co-authors have any competing interests.

**Acknowledgments.** We thank Mrs. Lidia Völker (ZALF) for help with data extraction from the GIS polygons.

**Financial support.** The work was funded by the German Research Foundation (DFG), grant number SCHA 1822/12-1 to Jörg Schaller. J. Strauss work was embedded into the 'Changing Arctic Ocean (CAO)' program (CACOON project [#03F0806A (BMBF)]. Grant NSF-1417700 to SMN.

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
