# Peer review of "Pan-Arctic soil element bioavailability estimations"

_Earth System Science Data, 2022_

## Author Response (AR1)

Comment on essd-2022-123
Anonymous Referee #1

Referee comment on "Pan-Arctic soil element availability estimations" by Peter Stimmler et al., Earth Syst. Sci. Data Discuss., https://doi.org/10.5194/essd-2022-123-RC1, 2022

Referee comment (RC): The manuscript "Pan-Arctic soil element availability estimations" by P. Stimmler presents Pan-Arctic concentration maps for amorphous silica, available Si, available Ca, available Fe, available P, and available Al based on selective extractions from 574 Arctic soils. I do appreciate the effort from the authors to provide this first order estimate as I think this contribution on mineral element distribution in Pan-Arctic soils is very much needed but I would like to see the limits of the approach more clearly stated. The dataset generated from the selective extractions on soils is certainly to report in ESSD. The concentration maps derived from the extractions are based on several hypotheses. Before publication, the manuscript should clarify some of the hypotheses, bring nuance in some of the conclusions, and clarify (especially in figures) that the concentration maps are for selectively extracted elements (available pool) to avoid any confusion or misuse of the data.

Author's response (AR):
The authors would like to thank the reviewer for this positive evaluation of our work and to evaluate that our dataset worth being published in ESSD. Based on your valuable feedback we will incorporate the suggested changes to the manuscript. Please find our detailed responses to the raised points below.
All line numbers revere to the marked-up version!

**All line numbers are those of the manuscript with traced changes.**

RC: General comments
1.  The concentration maps produced are for *available elements* (Si, Ca, Al, Fe, P) resulting from selective extractions (extracted by Mehlich III solution) and amorphous silica resulting from an alkaline extraction. For amorphous silica, this is clear that this is a portion of the Si pool. But for the available elements, this should be clarified throughout the entire manuscript to prevent from confusion with total element concentration, and especially in the title of the manuscript, in the title of the results section, in the text describing the data, and in the caption of the figure maps (Figure 4, 5, 6, 7, 8). This is mentioned in the method section but not enough in the data description, figures and conclusions. The suggestion is really to use a dedicated term for the element extracted by Mehlich III solution, such as "available Si, available Ca, available Fe, available P and available Al" throughout the text to discuss the data and to refer to these data in the figure caption. I suggest to clarify the "bioavailability" in the title of the manuscript (such as "First estimates of Pan-Arctic soil bioavailable element concentration maps"). This is to prevent a misuse of the data in the future or misunderstanding by the reader.

AR: We thank the reviewer for this helpful feedback.
We agree with your comments and will clarify and point out the element pool as bioavailable element concentrations in the text and in the figure legends to avoid confusion. Further, we will give a clearer definition of "bioavailable element concentrations".
We know about the given limitations of the extrapolation of our data and will point out them more clearly. Fort this we wrote a separate paragraph about the limitations of data and statistics.

RC:
2.  The approach of the authors is to use the *geological map* to extrapolate their data and create the maps (L91-92). Using the geological map to extrapolate selective extractions from soils requires the following hypothesis to be true: "that the soil parent material is the bedrock located underneath". This hypothesis should be discussed and more nuances should be given to this point especially in the Arctic. Indeed, abrasion by glaciers and transport of moraines by glaciers have generated and redistributed unconsolidated material (with mixed lithologies) at the surface in some Arctic regions. And in these regions, the soils are derived from this unconsolidated material integrating mixed lithologies (with a composition which may differ from the geological bedrock underneath). See the following publication https://doi.org/10.1130/G38626.1. Given the implication for the creation of the concentration maps, this is a very important point to discuss in the text to explain the limit of the approach.

AR: The information on the lithology of the underlying bedrock given in the Geological Map of the Arctic (Harrison et al. 2011) differs between mafic and sedimentary origin of the bedrock. Especially in Yedoma affected regions, and here especially in northern Siberia, the Lithology includes different types of sedimentary rocks (lithology 9: Sandstone, siltstone, shale, limestone). We will point out in the text, that the underlying bedrock origins from Pleistocene depositions in these regions and refer to (Kokelj und Burn 2005) for the potential of element release in these areas. We will further extend our discussion with this new material that clearly point out that the use of a geologic map for extrapolation of soil properties comes with certain limitations. (L558-L563)

RC:
3. The dataset presented comprises amorphous Si concentration and available Si concentration extracted by Mehlich III solution. The authors should clarify in which form is Si extracted by Mehlich III solution, and discuss the influence of the different forms of Si on P.

AR: Si extracted by Mehlich III is the silicic acid which is adsorbed to the soil particle surface and the free silicic acid. The description of this will be amended in the manuscript to clarify the differentiation. (L53-L60)

4. RC: The concept of "element availability or bio-availability" should be more clearly explained in the introduction. It is mentioned L103 that this is about "biological available element concentration". This same term should be used throughout the entire manuscript to avoid confusion.

AR: Thank you, we will define the term "bio-availability" in the context of the used extraction methods and use this term for the entire manuscript as suggested.

RC: Specific comments:
L58-59: the sentence compares sediments and soils. Do you mean Arctic soils derived from these sediments? The sentence and the references should be clarified.
AR: We refer to soils derived from the sediment deposits here. This may be fluvial depositions from nearby mountains. We will point this out more clearly in the text. (L72-71)

RC: L64-65: The influence of plant cycling and external inputs should also be considered.
AR: We appreciate with this feedback and will point out the importance of plants as a driver in the biogeochemical cycle, also the plant communities that are typical for acidic and non-acidic tundra (Walker et al. 2001). In the Yedoma regions we will point out the importance of current fluvial depositions (Monhonval et al. 2021). (L74-L79)

RC: L89: the impact of vegetation on mineral availability in soils is well demonstrated (Jobbagy and Jackson, Biogeochemistry 53: 51–77, 2001). In the Arctic, vegetation shift are suspected to influence element availability (e.g.,
https://doi.org/10.5194/bg-19-2333-2022, and also https://doi.org/10.1016/j.geoderma.2022.115915)
AR: We will point out the effect of plants on the accumulation of relevant element nutrients like P in the upper soil layer as discussed by (Jobbágy und Jackson 2001). Further, we will point out the interaction of element release and vegetation community in a warming Arctic as discussed in (Mauclet et al. 2022) and (Villani et al. 2022), a fact that certainly needs to be considered when estimating future effects of nutrient limitations on Arctic carbon cycle processes. (L106-L113)

RC: L96: "limiting nutrients for CO2 binding" the sentence should be revised
AR: We agree. We will modify this as recommended to:
"In addition, these elements, once transported to the marine systems, are limiting nutrients for $CO_2$ binding in terms of primary production by diatoms and coccolithophores." (L121)

RC: L118: the samples were oven dried at what temperature? Could you discuss the potential influence on the selective extractions (Mehlich III solution and De Master).
AR: The samples were freeze dried for 48h. We will point out the potential influence of freeze drying on element concentration in Mehlich III extracts and DeMaster. The use of dried soil material is standard for those extracts, again a point that we included in the methods description. (L148-L150

RC: L127: the amorphous silica extraction should be presented in a different paragraph. The term "available amorphous silica" is used. Could you clarify the availability of this pool given that this is a solid pool of amorphous silica. Can you explain how the De Master extraction technique for ASI used in the range of Arctic soils can be compared: can you discuss if there is a matrix effect? Can you explain the difference with the Si extracted by Mehlich III solution.
AR: We will explain the Mehlich III and alkaline extraction in two different paragraphs.
The Mehlich III is extracting the silicic acid which is adsorbed to the soil particle surface and the free silicic acid. The Alkaline extraction by DeMaster extracts the solid amorphous silicon of the soils. We will also clarify to the term "amorphous silicon" instead of "available amorphous silica" (L154-L160)

RC: L160: The use of word Era is inappropriate. Intervals of geological time scale are given formal names according to their length. There are four Eras: Precambrian, Paleozoic, Mesozoic, and Cenozoic. This should be revised.
AR: We will revise the word Era with "Period" as given in the geological map of the Arctic "Text-based formal geochronological unit lower in rank than era and higher than epoch, based on terms supplied by the ICS time scale (International Commission on Stratigraphy, 2009) (Harrison et al. 2011)" (L203)

RC: L208 and L210 are saying the opposite. Can you clarify.
AR: The Limestone of the Lithology Class 4 includes Shale, which can be a source of silicon minerals with high solubility. The Limestone of Class3 only contains low concentrations of Si containing minerals with a low solubility. A further analysis of the mineral composition was not done in by (Harrison et al. 2011) (L256-L257)

RC: L212 and L289: it should be mentioned clearly that this is fully expected for Ca to be the highest in the Limestone.
AR: We agree and will point out the expectation that under acidic extraction conditions limestone shows a high solubility, and by this the highest Ca concentrations. (L257+L347-348)

RC: L247-248: The sentence should be "Here, basalt and gneiss are dominant". Because basalt is always mafic, and gneiss is always metamorphic. If the aim of the authors was to insist on the fact that igneous and metamorphic rocks were dominant, the sentence could be "Here, igneous (lithological class 1; basalt) and metamorphic (lithological class 2; gneiss) are dominant.
AR: Thank you, we will modify the sentence as recommended. (L297)

RC: L269: here "available Si" should have been clearly defined earlier in the manuscript and well distinguished from the term "available amorphous Si". The differences between the two Si forms should be clarified. The term "available Si" should be used in the title of section 3.3.2. The same applies for section 3.3.3 for "available Ca" etc for the other sub- sections of the results.
AR: We will use the term "available" for all element data extracted by Mehlich III to make clear that the data is not about the total pool. Further, we will refer the alkaline extractable Si to "amorphous silicon". (L321)

RC: L364: a reference if needed. For complex or variable, do you mean "heterogeneous"?
AR: We will refer to "heterogeneous", as the fluvial deposits consist from different rock sources. As reference we will use (Schirrmeister et al. 2011) (L430-L431)

RC: L365: the sentence should be revised because I do not see what the authors mean by the parent material of the sedimentary rocks. Sedimentary rocks can be the parent material for a soil. And sediments of diverse origin can contribute to form a sedimentary rock.
AR: Thank you for pointing to this lack of clarity. We wanted to point out what you have written in your last sentence. We will modify the sentence to: Sedimentary rocks cover a wide range of pH, as these sedimentary rocks origin from diverse sediments. (L430-L431)

RC: L370: the sentence should be revised to bring more nuance because I do not see in the manuscript where it is "shown that changes in element availability will occur during permafrost thaw".
AR: We refer to a mobilization of elements from the current permafrost layer with deepening of the active layer due to thaw by warming. Further, run-off from thermokarst affected areas can modify elemental availability. We will make this clearer in the manuscript. (L438-L439)

RC: L371: this part of the sentence is not necessary as it describes what will come in section 4.4.

AR: we agree and will delete this part (L443-L448)

RC: L385-386: See earlier comment - the impact of vegetation on mineral availability in soils is well demonstrated (Jobbagy and Jackson, Biogeochemistry 53: 51–77, 2001). In the Arctic, vegetation shift are suspected to influence element availability (e.g., https://doi.org/10.5194/bg-19-2333-2022, and also https://doi.org/10.1016/j.geoderma.2022.115915)
AR: We will discuss the impact and limitation of vegetation on soil element availability including your suggested references. Please see response to comment above. (L454-L455)

RC: L387: a reference is needed to support the statement that "the vegetation involved in forming the soil ASi pool may be different", and the sentence should be clarified to explain "different than what".
AR: The Arctic vegetation and by this the processes of element cycling in soils have changed over the millennia (Andreev Andrei A. et al. 2003; ANDREEV et al. 2016; Schweger 1982) This hardens estimations of pools based on recent vegetation. (L458)

RC: L392: "higher ASi concentrations" than what? The authors should clarify if this ASi is available, what is the form of this Si, and how this form can compete for binding sites for P.
AR: Absolutely right, these clarifications have been added to the manuscript text. The ASi concentrations in the permafrost is higher compared to the concentrations of the Active Layer. ASi is a fraction of the particulate Si phases in soils. ASi is a main source of silicic acid which is able to bind on iron mineral phases and outcompete P from the surface binding sites (Schaller et al. 2019; Hömberg et al. 2020), increasing the concentration of available P. (L464)

RC:L396 until L411: this paragraph is presenting a large amount of information but the link with the presented data is not always clear (as explain here below for several examples). The objective of this ESSD manuscript is to report the selective extraction data, and to provide a first attempt to produce concentration maps for selected available elements.
This is a valuable contribution. And no further speculation or over-interpretation of the dataset should be included in the discussion.
AR: We agree. We will shorten the part of the discussion on potential interactions of the available Ca. (L443-L448)

RC: L399-400: This sentence should be revised or removed. It should be clarified if the authors refer to calcium carbonate dissolution (weathering) consuming CO2, or calcium carbonate formation releasing CO2.
AR: Thank you. In this case, we discuss the $CaCO_3$ neo-formation from free Ca in soils solution and $CO_2$. This was clarified in the revised text. (L473)

RC: L401: a reference is needed for this statement about Ca in Yedoma.
AR: we agree and we will use (Monhonval et al. 2021) here (L475)

RC: L401-402: the authors mention that "Ca concentrations are usually lower in deeper than in upper layers". A reference is needed. The authors should also consider the work from Kokelj and Burn, Can. J. Earth Sci. 42: 37–48 (2005), and several other papers by the same authors showing increase in cations in deeper layers depending on thaw history.
This sentence should be revised and discussed with the available data.
AR: We will revise the sentence to: In deeper soil layers, the data presented in Fig. S6 shows Ca concentrations are usually lower than in upper soil layers, which is in accordance to other studies (Kokelj und Burn 2005).
(L477)
RC: L403: "widespread decrease in Ca concentrations" in what? In soils? In sediments at depth? In selective extracts from soils? This should be clarified.
AR: this phrase was modified to: widespread decrease in available Ca concentrations in average at 0-1m depth (L479)

RC: L407: "amorphous aluminosilicates that mineralizes slowly"? the authors mean "crystallize slowly"? This should be verified because I do not see how an aluminosilicate can mineralize.
AR: We agree. The phrase was modified as recommended to "crystallize slowly" (L483)

RC: L407-408: the sentence about the role of Al on cytotoxicity is a statement that should be connected to the form of Al discussed in the manuscript. Is the Al extracted by Mehlich III solution a form of Al comparable to the form of Al cytotoxic? And can the authors clarify "OM respiration": respiration, or OM decomposition, respiration by microorganisms.
AR: The Mehlich III extracts of Al contains all soluble forms of $Al(OH)(H_2O)$ that are bioavailable for organisms, with Al being toxic. This additional information will be included in the specific text. (L484)

RC: L422 (and elsewhere in the manuscript): (higher available Ca and available Fe concentration in permafrost layer).
AR: We will add this as suggested (L500)

RC: L427-428: it should be mentioned that this is certain conditions
AR: earlier studies have shown this effect under slight acidic to alkaline and also from oxic to anoxic conditions (Schaller et al. 2019; Hömberg et al. 2020). Hence, we do not see why we should specify the conditions here. (L505-L506)

RC: L441: what is "free Si"? what form?
AR: Modified to: Soluble available forms of silicic acid and its polymers. (L504)

RC: L453: Increased "available" Fe concentration, in what? In soils? In solution?
AR: Increased available Fe concentrations at the depth of 0-1m. (L534)

RC: L454-455: the authors should better explain the second part of the sentence, "may only occur in the Canadian Shield". In soils developed on the Canadian Shield. Why? Here and Line 464, the generalties about these large areas should be nuanced, by saying "according to the sites studied within these lithological classes".
AR: we agree and will modify to: "can only be expected for soils of the Canadian Shield based on the data of Fig. S6, according to the sites studied within these lithological classes." (L534-L535)

RC: L453: Increased "available" Si concentration, in what? In soils? In solution?
AR: in the soil at depth of 0-1m. We will add this in the revised manuscript. (L542)

RC: L465-472: within this paragraph, the transfer of elements form soils to the sea is discussed. But processes during transfer should be accounted for and considered to nuance the discussion.
AR: We agree and will discuss the formation of precipitates during the transport. (L552-L554)

RC: L499: concentrations in available Si, Ca, Fe, P and Al between different Arctic regions.
AR: We agree. We will modify this as recommended. (L590)

RC: L500: the statement "the availability of the elements will change" should be removed, and replaced by "the exposed pool of the elements will change which will likely lead to difference in their availability according to our data."
AR: We agree. We will change this accordingly. (L592-L593)

RC: Table 1 caption: can you revise the caption and provide a general description before the column description. For L173-174, the last sentence of the caption is unclear.
AR: We agree and modify as recommended. We will include following sentence to clarify this: "The represented area is the share of the entire area from the Geological map of the Arctic presented in this study for the listed parameters" (L217-L219)

RC: Figure 1 caption: Can you revise the first sentence of the caption as this is not a "Map of extrapolated element concentrations" but these are lithologies.
AR: We agree. We will modify this to: "Map of represented lithologies" (L224)

RC: Figure 2: the X-axis is unclear. It should be clarified that class 1 is igneous, class 2 is metamorphic and the rest is sedimentary, or sedimentary and mixed.
AR: We will change the x-axis text to: "Lithological classes " and the  figure caption to: "Each color represents a bedrock lithology. Igneous: 1 […], Metamorphic : 2 […]. Sedimentary: 3 […]"(L278-L279)

RC: All captions from Figure 4, 5, 6, 7, 8 should be clarified to mention that this represents concentrations

in available Si, available Ca, available Fe, available P and available Al (extracted by Mehlich III solution). This is essential to avoid confusion or misuse of the data in the future.
AR: Thank you. We will modify this in the manuscript to e.g. "Map of mean concentration of…" (L341)

RC: Figure S4: the caption is not in the same standard than the other captions. Arctic with capital A.
AR: Thank you very much for this hint. We will modify this caption to the same standard as the other captions.

RC: Figure S6: clarify that it is considered that organic layer and mineral layer are part of the active layer.
AR: We will modify this as recommended

**RC: Technical corrections**
L71: these elements
AR: We will change this as recommended. (L119)

RC: L73: here and throughout the entire manuscript, all chemical formula should be with numbers in indices ($Si(OH)_4$). This is true for many places in the document, for $CO_2$, etc…
AR: We will check this and add this if needed in the whole manuscript. (L121)

RC: L76: binds
AR: We will change this as recommended. (L92)

RC: L124-125: "The analysis was done by" should be replaced by "The concentration in Si, Ca, Fe, Al and P was measured by inductively coupled……"
AR: We will change this as recommended. (L163)

RC: L151: using Equation 1
AR: We will change this as recommended. (L194)

RC: L396: Calcium instead of Ca to avoid starting the sentence with a chemical symbol.
AR: We will change this as recommended. (L473)

RC: L429: emissions
AR: We will change this as recommended. (L507)

RC: L436-437: in some conditions
AR: We will change this as recommended. (L515)

RC: L441: seems to limit
AR: We will change this as recommended. (L521)

RC: Figure S3. Arctic with capital A
AR: We will change this as recommended.

RC: Figure S7: should be Phosphorus (not Phosphor)
AR: We will change this as recommended.

**Literaturverzeichnis**

ANDREEV, ANDREI A.; TARASOV, PAVEL E.; Wennrich, Volker; MELLES, MARTIN (2016): Millennial-scale vegetation changes in the north-eastern Russian Arctic during the Pliocene/Pleistocene transition (2.7–2.5 Ma) inferred from the pollen record of Lake El'gygytgyn. In: *Quaternary Science Reviews* 147, S. 245–258. DOI: 10.1016/j.quascirev.2016.03.030.

Andreev Andrei A.; TARASOV, PAVEL E.; ARASOV, PAVEL E.; Siegert, Christine; EBEL, TOBIAS; KLIMANOV, VLADIMIR A. et al. (2003): Late Pleistocene and Holocene vegetation and climate on the northern Taymyr Peninsula, Arctic Russia. In: *Boreas* 32 (3), S. 484–505. DOI: 10.1080/03009480310003388.

Harrison, J. C.; St-Onge, M. R.; Petrov, O. V.; Strelnikov, S. I.; Lopatin, B. G.; Wilson, F. H. et al. (2011): Geological map of the Arctic. Online verfügbar unter https://geoscan.nrcan.gc.ca/starweb/geoscan/servlet.starweb?path=geoscan/downloade.web&search1=R=287868 , zuletzt geprüft am 27.08.2019.

Hömberg, Annkathrin; Obst, Martin; Knorr, Klaus-Holger; Kalbitz, Karsten; Schaller, Jörg (2020): Increased silicon concentration in fen peat leads to a release of iron and phosphate and changes in the composition of dissolved organic matter. In: *Geoderma* 374, S. 114422. DOI: 10.1016/j.geoderma.2020.114422.

Jobbágy, E.; Jackson, Robert B. (2001): The distribution of soil nutrients with depth: Global patterns and the imprint of plants. In: *Biogeochemistry* (53), S. 51–77. Online verfügbar unter https://link.springer.com/article/10.1023/A:1010760720215, zuletzt geprüft am 17.01.2023.

Kokelj, S. V.; Burn, C. R. (2005): Geochemistry of the active layer and near-surface permafrost, Mackenzie delta region, Northwest Territories, Canada. In: *Can. J. Earth Sci.* 42 (1), S. 37–48. DOI: 10.1139/e04-089.

Mauclet, Elisabeth; Agnan, Yannick; Hirst, Catherine; Monhonval, Arthur; Pereira, Benoît; Vandeuren, Aubry et al. (2022): Changing sub-Arctic tundra vegetation upon permafrost degradation: impact on foliar mineral element cycling. In: *Biogeosciences* 19 (9), S. 2333–2351. DOI: 10.5194/bg-19-2333-2022.

Monhonval, Arthur; Mauclet, Elisabeth; Pereira, Benoît; Vandeuren, Aubry; Strauss, Jens; Grosse, Guido et al. (2021): Mineral Element Stocks in the Yedoma Domain: A Novel Method Applied to Ice-Rich Permafrost Regions. In: *Front. Earth Sci.* 9, Artikel 703304. DOI: 10.3389/feart.2021.703304.

Schaller, Jörg; Faucherre, Samuel; Joss, Hanna; Obst, Martin; Goeckede, Mathias; Planer-Friedrich, Britta et al. (2019): Silicon increases the phosphorus availability of Arctic soils. In: *Scientific reports* 9 (1), S. 449. DOI: 10.1038/s41598-018-37104-6.

Schirrmeister, L.; Kunitsky, V.; Grosse, G.; Wetterich, S.; Meyer, H.; Schwamborn, G. et al. (2011): Sedimentary characteristics and origin of the Late Pleistocene Ice Complex on north-east Siberian Arctic coastal lowlands and islands – A review. In: *Quaternary International* 241 (1-2), S. 3–25. DOI: 10.1016/j.quaint.2010.04.004.

Schweger, Charles E. (1982): LATE PLEISTOCENE VEGETATION OF EASTERN BERINGIA: POLLEN ANALYSIS OF DATED ALLUVIUM. In: Paleoecology of Beringia: Elsevier, S. 95–112.

Villani, Maëlle; Mauclet, Elisabeth; Agnan, Yannick; Druel, Arsène; Jasinski, Briana; Taylor, Meghan et al. (2022): Mineral element recycling in topsoil following permafrost degradation and a vegetation shift in sub-Arctic tundra. In: *Geoderma* 421, S. 115915. DOI: 10.1016/j.geoderma.2022.115915.

Walker, D.A; Bockheim, J.G; Chapin III, F.S; Eugster, W.; Nelson, F.E; Ping, C.L (2001): Calcium-rich tundra, wildlife, and the "Mammoth Steppe". In: *Quaternary Science Reviews* 20 (1-3), S. 149–163. DOI: 10.1016/S0277-3791(00)00126-8.

**Comment on essd-2022-123**

Anonymous Referee #2

Referee comment on "Pan-Arctic soil element availability estimations" by Peter Stimmler et al., Earth Syst. Sci. Data Discuss., https://doi.org/10.5194/essd-2022-123-RC2, 2023

Referee comment (RC): General comments

1) The manuscript entitled "Pan-Arctic soil element availability estimations" aims to predict bioavailable silica, silicon, phosphorous, calcium, iron, and aluminum concentrations. It is based on a precious measured dataset and applies a lithology related spatial interpretation approach. The reviewer is fascinated by the efforts paid for the sampling, measurements, and database construction. This part, without doubt, is valuable to be published in ESSD. The applied model compilation for regional map construction is simple and useful, even though it simplifies the question and must be handled with limitations. The paper in its present form is not ready for publication, but definitely promising. After clarifying some minor points, it would be a relevant contribution.

Authors' response (AR): Thank you very much for your valuable comments and positive evaluation of our manuscript.

**All line numbers are those of the manuscript with traced changes.**

RC:

2) I would be glad to get familiar with the sampling in detail for the dataset construction. Nothing is given, for example, regarding the duration. Was it within one season, one year, or one decade? What if seasonal variability of the available elements may be high due to Eh, pH changes, or increased biological activity among sampling dates? Was there repeated sampling on particular sites? If so, is there a temporal difference between the results? Or available results to compare from the literature?

AR: The samples were taken by the co-Authors during one decade, due to the high logistic effort of sampling in the Arctic. Some ecological variabilities cannot be excluded, but we assume high stability of the conditions in the Active Layer and especially in the permafrost layer. For some locations sampling were repeated spatially. A sampling regarding temporal differences was done in none of the locations. The samples are not comparable with previous studies as so far no study exists showing comparable data. For clarifying the sampling we added a short paragraph to the method section.

RC:

3) However, my main concern is about the graphical interpretation. Based on the applied approach, mean bioavailable concentrations for each lithological class and depth are shown. I am unsure if an average is acceptable for such huge areas with high variability. E.g., if variability exceeds a certain threshold, that category is presumed to be diverse and cannot be described with a single average. This point has to be taken into account. Being more general, the accuracy and the limitations of the maps are not discussed. Even the simplest model has to be validated.

AR: We are aware about the difficulties and uncertainties of extrapolating our dataset on such a large scale. Nonetheless, our dataset is much larger than those used in studies before. But of course we absolutely agree with your statement that even the simplest model has to be validated. To reduce statistic uncertainties we did bootstrapping to produce reliable results and to discuss the presented data with standard deviations. This technique was already used by Hugelius et al. 2014, Strauss et al 2013 and 2017 estimating the carbon pools of the Arctic, now being a well-used basis for modelling studies. As our data present available element concentrations and not total element pools the validation with external datasets is not possible, as no other study has shown such data for this area. Nevertheless, in the revised version of the manuscript we will point more clearly to lithologies with a high assumed variability in the data according to Table S4.

RC:

4) The high variance of available element concentrations within a lithological unit, among others, may indicate the role of geomorphology. Earth surface processes may rework the active (soil?) layer even under Arctic conditions resulting in lithology independent conditions and causing outliers in the database. Accordingly, the limitations of the applied approach need to be emphasized.

AR: We are aware about the processes that can modify element concentrations within a lithology class. Weathering, geomorphology, crystallinity, climate and vegetation can have an impact on the availability of the elements. Our approach is to use the element availability under the environmental conditions as given and consider this effects by bootstrapping. We will discuss this effects and the limitations of our dataset in the manuscript comprehensively, also emphasizing the weaknesses/limitations of the approach.(L558-L563)

RC: Specific comments

5) Authors on many occasions (practically all along the paper) are talking simply about "element concentrations," even though these are supposed to mean "available concentrations."

AR: Thank you for this comment. We will change the text to "available element concentrations" as suggested

RC:

6) Most maps are provided with a continuous color scale suggesting continuous values. In contrast, the map is of discrete values. Therefore, I think legends such as those in Fig1 would fit better.

AR: The presented data belongs to different categories. The map presented in Figure 1 shows lithologies (nominal data). To avoid the assumption that lithologies follow linear correlations we used distinct data, similar to those regarding to lithologies in the Geological Map of the Arctic. Maps presented in Fig.3-Fig.8 show the range of available element concentrations (numeric data). Here, distinct data could not represent the linear nature of the data or the fine scaling. In this case the continuous data increase data quality and simplify interpretation of the graph.

RC:

7) It would be nice to somehow compare the present results to those presented in the literature. The stock, at least for a single layer, can be roughly estimated using bulk densities derived from the core volumes and the weights.

AR: This is an interesting suggestion and would also follow the approach done by Hugelius et al or Strauss et al. For this paper we decidet not going this way but leave this for a following manuscript (e.g. as this would double the presented data and figures).

RC: l.38 I am not sure if "frozen ground" is the best synonym for "permafrost soil." The term "permafrost" is believed to highlight the continuum.

AR: We will change the term to "permafrost" in the text (L41)

RC: l.43 Do you mean organic carbon?

AR: Yes, organic carbon, changed accordingly (L45)

RC: l.66 I would also mention the role of SOM here beyond the minerals.

AR: We can discuss the role of SOM in the fitting paragraph. For a better understanding we will focus on just elements in this paragraph.

RC: l.151 vs. 154 Please use the same term for the equation/formula

AR: Yes, thank you, we will change this in the manuscript. (L194)

RC: l.299 "slightly lower" I am not sure if these differences are significant. However, from a purely statistical point of view, if the difference is insignificant, there is no difference at all.

AR: You are absolutely right. In this case we describe a dataset as requested by ESSD. Further studies using this dataset should do the statistical analysis.

RC: l.317-321. Please clarify.

AR: modified to: Fe concentrations (Mehlich III extractable) were higher in the Canadian Arctic, compared to the Siberian Arctic (Fig. 6).

RC: l.327 I probably missed something, but this low SD contradicts the boxplot size presented in fig2. Which one is right?

AR: The boxes are correct, the data is 2.52± 0.7 mg/g DW Al. We will correct this in the text. (L387)

RC: l.374-380 Some parts of the discussion would fit better with the introduction as it repeats statements that are independent of the results

AR: We agree and will change this in the manuscript. (L443-L448)

**RC: Technical corrections**

l.39 Here, and in many other cases, the lower/upper case indexes are missing

AR: We will check the manuscript for this and adapt this as suggested.

RC: l.71 these

AR: We will change this accordingly. (L72)

RC: l.154 In eq.1, "m" is given instead of "cm."

AR: We will change this in the equation. (L197)

RC: l.252-253

AR: We will delete this. (L302)

RC: l.345 Please clarify.

AR: Mobilization of Al stored in the permafrost due to deeper thawing depth and cryoturbation.

RC: Fig.1 Hard to distinguish among the applied colors.

AR: We used similar colors to those used in the Geological Map of the Arctic, to avoid confusion with the regular color code.